# Review article: Current approaches and critical issues in multi-risk recovery planning of urban areas exposed to natural hazards

Soheil Mohammadi[1], Silvia De Angeli[1], Giorgio Boni[1], Francesca Pirlone[1], Serena Cattari[1]

[1] Department of Civil, Chemical and Environmental Engineering, University of Genoa, Genova, 16145, Italy

*Correspondence to*: Soheil Mohammadi (soheil.mohammadi@edu.unige.it)

**Abstract.** Post-disaster recovery has been addressed in the literature by different sectoral perspectives and scientific communities. Nevertheless, studies providing holistic approaches to recovery, integrating reconstruction procedures and socio-economic impacts, are still lacking. Additionally, there is a gap in disaster recovery research in addressing the additional challenges posed by complex, multiple, and interacting risks acting on highly interconnected urban areas. Furthermore, recovery has only been marginally explored from a pre-disaster perspective in terms of planning and actions to increase urban resilience and recoverability.

This paper provides a critical review of existing literature and guidelines on multi-risk disaster recovery with the twofold aim of identifying current gaps and providing the layout to address multi-risk recovery planning tools for decision-making. The literature on disaster recovery is investigated in the paper by focusing on: the definition of the recovery phase and its separation or overlapping with other disaster risk management phases; the different destinations and goals that an urban system follows through recovery pathways; the requirements to implement a holistic resilience-based recovery roadmap; the challenges for shifting from single-risk to multi-risk recovery approaches; and the available tools for optimal decision-making in the recovery planning. Finally, the current challenges in multi-risk recovery planning are summarised and discussed. This review can be a ground basis for new research directions in the field of multi-risk recovery planning to help stakeholders in decision-making and optimise their pre-disaster investments to improve the urban system's recoverability.

## 1    Introduction

The frequency of natural hazards and consequent disaster events has increased in recent decades (CRED, 2022). Moreover, the extent of these events' impact, in terms of both the economy and humankind, has shown exponential growth (Cerè et al., 2017). Additionally, by 2050, urban areas would house almost 70% of the world's population, making them global centres of human settlement and capital accumulation and the locations most exposed to natural hazard events (Ritchie and Roser, 2018). To cope with natural hazard event impacts, planning and procedures that anticipate their occurrence, mitigate possible damages, and allow for the speedy restoration of key services and recovery are required (Berke et al., 2009).

In the last decades, the paradigm of disaster risk management has shifted progressively from focusing mainly on increasing the resistance against hazards to resilience planning, with the goal of reducing direct and indirect impacts in communities facing natural hazards and improving their recoverability (Leichenko, 2011). The origin of the modern resilience theory and its application to natural ecosystems back to Holling's seminal work in 1973 (Holling, 1973). Holling used the resilience term to describe the system's capability to remain functional and "persist" after changes that might happen. Holling's writing was the basis for developing the resilience concept and applying it to understand the performance of complex systems, i.e. systems "in which there are multiple interactions between many different components" (Rind, 1999), when encountering disturbances (Walker et al., 2004; Holling et al., 2002). According to an ecological science perspective, disturbances can be seen as massively destructive and rare events (Rykiel Jr, 1985) that impact the system from the outside. Folke (2006) strives for the elaboration of the concept and provides the resilience theory not just restricted to the extent of a system characteristic but as a mindset. Over time, the comprehensive resilience concept has been extended to various fields and domains, including natural hazards and risk management (Coaffee, 2008; Cutter et al., 2008; Klein et al., 2003; Bruneau et al., 2003), climate change adaptation (Nelson et al., 2007; Tanner et al., 2009) and planning (Davoudi et al., 2012; Wilkinson, 2012).

Urban areas have been usually described as 'complex systems' (Brugmann, 2012; da Silva et al., 2012), composed of a series of interconnected social, ecological, and technical (or physical) networks through which services and goods are provided. Moreover, the interplay between people, activities, institutions, resources, and processes contribute to the complex, dynamic and unpredictable nature of urban systems. Historically, urban areas were often considered a safe refuge, shielding inhabitants from the adverse effects of natural hazards. However, a paradigm shift has occurred, acknowledging that urban areas are now recognized as focal points where disasters and risks converge (Šakić Trogrlić et al., 2018). In fact, the rapid, unplanned, and not risk-informed expansion of urban areas often necessitates construction in locations that are susceptible to multiple hazards (Cremen et al., 2023). This is primarily due to limited available land or insufficient time and resources to thoroughly evaluate these areas for their susceptibility to potential interactions between multiple hazards (Jenkins et al., 2023).

As one of the most complex systems exposed to natural hazards, urban areas have also been included in the resilience concept's extent. Urban resilience has been defined variously in the literature. One of the most comprehensive definitions is provided by Meerow et al. (2016): "Urban resilience refers to the ability of an urban system and all its constituent socio-ecological and socio-technical networks across temporal and spatial scales to maintain or rapidly return to desired functions in the face of a disturbance, to adapt to change, and to quickly transform systems that limit current or future adaptive capacity." The phrase "ability to return" stands out in this definition. The fundamental concern that arises, however, is how to ensure that the urban areas, as complex systems with various components and interconnected networks, have the ability to return or recover after a disaster occurs. On the other hand, the term "maintain" can be seen in Meerow et al. (2016) definition as a reference to the ability of a resilient urban system to preserve what currently exists, which is more closely tied to the concepts of disaster risk preparedness and response.

Enhancing resilience in urban areas can be a complex task, considering the multitude of components, processes, and interactions occurring within and beyond the physical, legal, and virtual boundaries of the urban area (Desouza and Flanery,

2013). An analysis of previous international disaster responses reveals a preference among international humanitarian agencies to provide assistance in rural areas when disasters impact both rural and urban regions (MacRae and Hodgkin, 2016) and when

it comes to recovery, urban reconstruction efforts have primarily been undertaken within the scope of national reconstruction programs, receiving limited support from international humanitarian agencies because of the complexities of the recovery process in urban areas (Daly et al., 2017). Urban rebuilding faces a key challenge in dealing with the intricate network of stakeholders and the complex urban settings due to multiple governance layers, diverse community interests, and a mix of private and public entities. This complexity makes coordination and decision-making significantly more complex than in rural

areas (Daly et al., 2017).

The resilience concept can be reflected in preparedness, response, recovery, and adaptation actions, depending on the temporal domain. Even though numerous studies have quantified the benefits of investing in preparedness by comparing potential damage and preparedness costs (Goldschmidt and Kumar, 2019; David R. et al., 2009; Kousky et al., 2019; Heo and Heo, 2019), there is still a notable absence of research that systematically evaluates the relationships and implications of different

actions on one another in resilience-based urban planning (Rus et al., 2018). This lack in the literature is reflected also in the confusion felt by stakeholders involved in the resilient urban planning process. Indeed, decision-makers do not know how to choose an investment direction among the various phases of the Disaster Risk Management (DRM) cycle (Kawasaki and Rhyner, 2018). Are they supposed to protect and strengthen the system, work to make it more recoverable, or even set aside funds for disaster recovery? Although the resilience of an urban system is not solely determined by its ability to restore from

a disruptive event, the recovery process represents one of the most critical and significant aspects contributing to the overall system resilience. To corroborate this perspective, Manyena et al. (2019) indicate that, since 1980 the terms "return to equilibrium," "bounce-back," "recover," "restore," "bounce-forward," "rebound," "rebuild," and "reorganize", which all have the connotation of recovery, have been frequently used in the different resilience definitions provided in the literature. According to McEntire et al. (2002), the term "resilience" emerged as a reaction to or alternative for the term "resistance". The

key difference between these two concepts is that contrary to the resistance idea that prevention is the main strategy, natural hazard events inevitably occur in resilience discourse and disaster avoidance is not always achievable. As a result, the main issue in resilience is the need to focus on recovering from disasters as quickly and effectively as feasible (McEntire et al., 2002). Indeed, different researchers have used indicators such as the system's ability to function during recovery, the speed of recovery, the quality of recovery, and the area under the recovery curve to measure the system's resilience (Bruneau et al.,

2003; Rus et al., 2018; Soltani-Sobh et al., 2016; Zhang and Wang, 2016). Although recovery is a fundamental aspect that is well captured by the resilience concept definition, this DRM phase is the least explored in the literature in comparison to the others (Der Sarkissian et al., 2021; Rodríguez et al., 2018).

When urban systems are affected by complex disaster scenarios, involving multiple hazards and potential impact interactions, addressing the recovery process can become more complicated. In that circumstance, the system must be resilient to many

types of risks, embracing a multi-risk perspective, which will introduce additional challenges in the decision-making process regarding the recovery (Curt, 2021). Multi-(hazard)-risk, as collectively named by Ward et al. (2022), encompasses all disaster

risk assessment and management approaches that consider interactions or interdependencies among different hazards, vulnerabilities, or risks. These approaches can better capture complex risk dynamics which are increasingly impacting urban areas worldwide. The interrelationship of multiple hazards and their impacts, as well as the implications of DRM decisions on

different economic sectors and regions, and the diverse impact of disaster risk reduction measures on different risks, challenge recovery in multi-hazard environments (Ward et al., 2022; Hochrainer-Stigler et al., 2023). Among the different types of multi-hazard interaction mechanisms that can lead to a disaster such as compound, consecutive, triggering, or cascading, (Gill and Malamud, 2014; Marzocchi et al., 2009; Tilloy et al., 2019), the occurrence of consecutive disasters is specifically challenging for the recovery process. Consecutive disasters are two or more disasters that occur in succession and whose direct impacts overlap

spatially before the recovery from the prior event is considered complete (de Ruiter et al., 2020). The results of the interaction between the impacts generated by two consecutive hazards depend on the time interval between them, the rate of recovery of the system or the asset, or a combination of them (De Angeli et al., 2022; Marzocchi et al., 2012). As a real-world example in the western part of Iran, a devastating earthquake of magnitude 7.3 $M_w$ occurred on November 12, 2017, at the Iran-Iraq border, causing the death of at least 630 people (Naserieh et al., 2022). The recovery efforts began by providing

temporary shelters to the more than ten thousand affected people (Omarzadeh et al., 2021). Before the earthquake, the country was facing a prolonged period of drought, and no one anticipated the possibility of a flood occurring within some months (Yadollahie, 2019). However, from mid-March to April 2019, widespread flash flooding occurred, affecting large areas of Iran, including the regions that were undergoing recovery from the earthquake (Miri et al., 2023). The potential occurrence of a flood was not considered, leading to the establishment of temporary shelters alongside canals, which resulted in the flooding

of the people residing in those shelters and imposed significant economic impacts on them.

This study analyses the existing disaster recovery literature and guidelines in the realm of natural hazards with the dual goals of (i) identifying current issues in multi-risk recovery planning for urban areas and (ii) laying the groundwork for developing multi-risk decision-making planning tools for recovery. The needs and present limits of building a multi-risk tool for recovery planning are addressed. The final goal is to propose new research directions that can inform stakeholders' decision-making

processes and help optimize their investments in the pre-disaster phase, contributing to the enhancement of urban areas' recoverability. It is crucial to emphasize that, while urban recovery planning encompasses various actions and aspects, in this research we focus on decision-making concerning investment prioritization to improve the resilience of physical elements (e.g., structures, buildings, infrastructures, open spaces, etc.) at the urban scale.

The paper is organised as follows: Sect. 2 explains the implemented methodology that allowed the identification of three main

research issues in multi-risk recovery planning, while each issue is extensively discussed in Sects. 3 (Issue 1), 4 (Issue 2), and 5 (Issue 3). Then, the current challenges in implementing multi-risk recovery planning resulting from the literature review are discussed in Sect. 6. The final outcomes of this critical literature review will set the basis to outline the new research directions and help stakeholders improve the urban system's recoverability (Sect. 7). The examined references, together with their critical classification (coherently to the one adopted for the critical review discussed in Sects. 3, 4 and 5), are reported in the

supplementary material of the paper to provide a comprehensive overview for the reader.

## 2    Methodology

In this study, we used a critical literature review (Grant and Booth, 2009; Snyder, 2019) approach. The literature on disaster recovery includes a broad group of researchers from several disciplines who have produced a large body of work. We sought a balanced and critical assessment of the literature (versus a systematic mapping of all relevant literature (Wong et al., 2013)), generating and responding to precise questions (Boaz et al., 2002). Specifically, the critical literature review focuses on natural hazards and aims to identify the most pressing challenges in multi-risk recovery so that they can be incorporated into the development of a multi-risk decision-making tool. The review has been guided by the following research questions:

- *Question 1.* What is the relationship between recovery and the other DRM cycle phases?
- *Question 2.* What is the final goal of the recovery process, and how can an increase in urban system resilience be ensured?
- *Question 3.* What are the most important physical prerequisites for the urban system to begin and sustain recovery in a multi-risk environment?
- *Question 4.* What are the methods and models currently used for prioritizing investments in physical elements to improve disaster recovery from natural hazards in urban areas?

The questions have been answered considering the following constraints and assumptions:

- *Recovery is analysed through the lens of resilience, considering specifically how disaster recovery can contribute to improving urban resilience.* Given that resilience is not just tied to recoverability, it is vital to understand how the recovery phase of DRM interacts with other phases. Considering this, Question 1 and Question 2 have been raised. Addressing these two questions would necessitate understanding both the phenomenology of the recovery as well as its role, beginning and end points within the DRM context.
- *The primary focus of this research is on urban physical assets, and how investment prioritization can increase their resilience from multiple hazards and facilitate the recovery process.* This is the core issue that should be considered when trying to answer the first two questions, which in turn would cause the second and third questions to emerge. More specifically, Question 3 addresses this issue of understanding how to address pre-disaster recovery in a multi-risk environment, and Question 4 is posed to understand and analyse the decision-making methods for investments in physical elements and the strengths and weaknesses of each method.

The review has included relevant publications found in Google Scholar and Scopus databases and the material available in the International Recovery Platform (UNDRR, 2022). To address the specific Research Questions, the following keywords were selected: difference between recovery and emergency, pre-disaster recovery planning (Question 1); resilient recovery, socio-economic aspects of the recovery (Question 2); recovery requirements, post-disaster needs, multi-risk recovery (Question 3); recovery optimization model, recovery planning (Question 4). These searches yielded approximately 250 papers/documents, of which we used roughly 130 to inform the findings in this paper. The remaining 130 papers have been selected for the following reasons:

- The recovery phase and its activities are the key topics and are addressed in detail.
- The publications address specifically only natural hazards or in combination with other types of hazards, such as technological or human-made hazards.
- Recovery is addressed at the urban scale.
- The publications focus on the physical elements and their function for the urban area's recovery (this has been considered specifically in answering research questions (iii) and (iv).

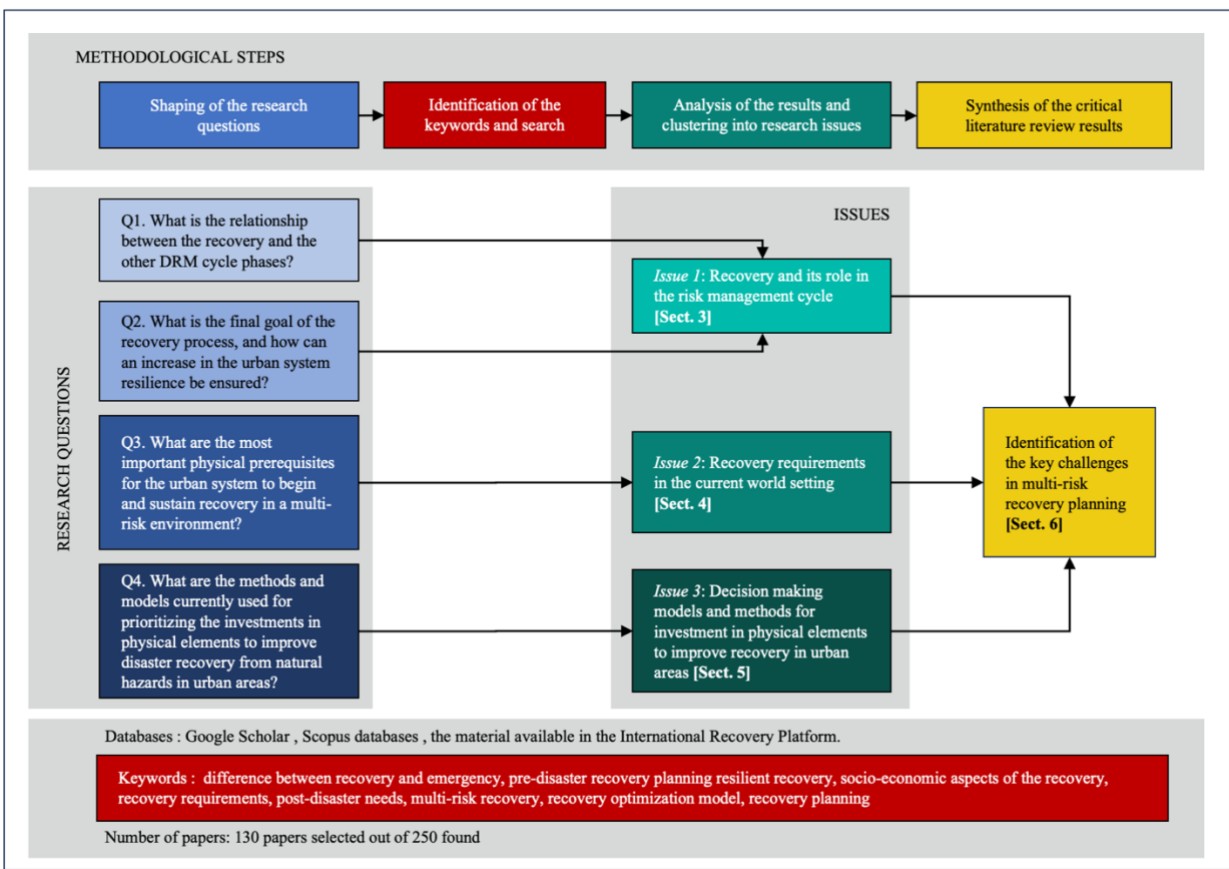

**Figure 1.** Graphical representation of the methodological steps implemented to perform the critical literature review on multi-risk recovery planning, highlighting the relationship between the guiding Research Questions and the multi-risk recovery planning Issues.

The obtained results have been clustered into a series of research issues, each one discussed in a separate session of the manuscript:

- *Issue 1*: Recovery and its role in the risk management cycle (Sect. 3) that corresponds to Questions 1 and 2.
- *Issue 2*: Single and multi-risk recovery requirements in the current world setting, (Sect. 4) that corresponds to Question *3*.

- *Issue 3*: Decision making models and methods for investment in physical elements to improve recovery in urban areas (Sect. 5) that corresponds to Question 4.

The outcomes of the critical literature review explained in detail in Sects. 3, 4, and 5, are then summarised in a series of key findings (Sect.6). The overview of the applied methodology as well as the relationship between the multi-risk recovery planning Issues and the guiding research questions are graphically depicted in Fig. 1.

## 3 Issue 1: Recovery and its role in the risk management cycle

### 3.1 What is recovery?

Despite disaster recovery having been studied from several perspectives, there was a lack of theory explaining recovery in the literature until the early twenty-first century (Chang, 2005). Early definitions of recovery included some pre-defined activities that should be completed in a timely order, with the result being a return to normalcy (Haas et al., 1977). Occasionally some progressive concepts were added, such as 'risk reduction' or 'decreased vulnerability' (Whyte, 1979). However, these first definitions completely neglected to consider socio-economic aspects affecting the complex recovery processes, and
oversimplified recovery as a well-defined and uniform path in all societies or even all scales of a single society (Holton, 2000; Sullivan, 2003).

A selection of definitions provided by different authors that illustrate the evolution of the recovery concept is given in Table 1 (the full table is provided in the supplementary material). Recovery, according to Nigg (1995), is more than just reconstructing the built environment, and the activities that shape this process can be influenced by pre- and post-disaster circumstances.
Smith and Wenger (2007) proposed a more holistic definition of recovery that considers the various groups' differential recovery steps and the socioeconomic aspects of the process while also emphasising the importance of pre-disaster planning. The definitions of recovery found in the literature can be separated into two groups (Winkworth, 2007; Ryan et al., 2016):

   i.   Definitions that focus on recovery as a desired outcome per se.
   ii.  Definitions that consider recovery as a process that leads to one or more desired outcomes.

In the second group, i.e., definitions that see recovery as a process, recovery refers to coordinated efforts in terms of decisions and actions to aid communities in returning to their pre-disaster state or even regenerating them to become less risky than before the disaster (Parker, 2006). Recovery, according to Mileti (1999), for example, "is not just a physical outcome but a social process that encompasses decision-making about restoration and reconstruction activities". All definitions reported in Table 1, except the one provided by Quarantelli (1989), conceptualize recovery as a process rather than a desired outcome.
In its terminology, the UNDRR (2020) distinguishes between 'reconstruction' and 'recovery'. The recovery definition is provided in the last line of Table 1. Reconstruction is defined as "the medium- and long-term rebuilding and sustainable restoration of resilient critical infrastructures, services, housing, facilities, and livelihoods required for the full functioning of a community, or a society affected by a disaster, aligning with the principles of sustainable development and 'Build Back Better' (BBB), to avoid or reduce future disaster risk." According to the UNDRR (2020), BBB is defined as "the use of the

recovery, rehabilitation, and reconstruction phases after a disaster to increase the resilience of nations and communities through integrating disaster risk reduction measures into the restoration of physical infrastructure and societal systems, and into the revitalization of livelihoods, economies, and the environment".

Table 1. Recovery definitions given by nine different sets of authors reported from the less to the most recent (the complete table is available in the supplementary material)

| Author(s) | Definition |
|---|---|
| Rubin and Barbee (1985) | "Long-term recovery (i.e., the reconstruction process) is characterised by attention to rebuilding and new construction; restoration of major urban services; and review of pre-disaster land uses, especially insofar as they include consideration of local hazards in the recovery plans for the affected area." |
| Quarantelli (1989) | "Disaster recovery implies that everything works out fine after the disaster." |
| Nigg (1995) | "Recovery is a social process that begins prior to disaster impact and encompasses decision making concerning restoration." |
| Australian Institute for Disaster Resilience (1998) | "The coordinated process of supporting disaster-affected communities in reconstruction of the physical infrastructure and restoration of emotional, social, economic and physical well-being." |
| Mileti (1999) | "Process of interaction and decision-making among a variety of groups and institutions, including households, organisations, businesses, the broader community and society." |
| Winkworth (2007) | "Recovery comes to signify an active process of integrating traumatic events associated with a disaster so that destructive impacts are minimised and so that individuals, communities and governments are able to move forward into a post-disaster future in which the world has changed." |
| Smith and Wenger (2007) | "The differential process of restoring, rebuilding, and reshaping the physical, social, economic, and natural environment through pre-event planning and post-event actions." |
| Smith et al. (2018) | "The differential process of restoring, rebuilding, and reshaping the physical, social, economic, and natural environment through pre-event planning and post-event actions that enhance the resilience and adaptive capacity of assistance networks to effectively address recovery needs that span rapid and slow-onset hazards and disasters." |
| UNDRR (2020) | "The restoring or improving of livelihoods and health, as well as economic, physical, social, cultural, and environmental assets, systems and activities, of a disaster-affected community or society, aligning with the principles of sustainable development and 'Build Back Better', to avoid or reduce future disaster risk." |


The fundamental distinction between the definitions of 'recovery' and 'reconstruction' provided by UNDRR (2020) lies under the fact that the reconstruction definition refers to more tangible elements. On the other hand, in the recovery definition a broader array of functions, (sub)systems, and activities of a community or society, which also encompass intangible elements, are mentioned. This emphasises the fact that recovery encompasses more than the rebuilding of the physical components of

the systems. In addition, while both definitions of reconstruction and recovery from UNDRR contain advice about aligning with BBB principles, the reconstruction definition refers to the goal of reaching a full community functioning, while the recovery definition emphasises more on the opportunity for 'improvement.' Therefore, by comparing the definitions, reconstruction can be regarded as a part of the recovery process, dividing all recovery activities into two primary domains:

  i.   Physical reconstruction.

ii.   Non-physical reconstruction-oriented activities, which we will refer to as 'socio-economic activities'.

The socio-economic activities encompass local business recovery efforts, community participation, building social connections, psychological support, activation of NGOs and voluntary groups, and institutionalisation, among others.

In this study we define functional recovery as a process addressing simultaneously physical and non-physical reconstruction activities whose primary purpose is the maintenance of community functions throughout and after recovery. Functional

recovery would be impossible if the focus was solely on physical reconstruction while the socioeconomic aspects of recovery were ignored. The importance of considering socio-economic recovery and its relationship with physical reconstruction and their impact on each other will be discussed in greater depth in Sect. 4, based on literature. The functional recovery zone is depicted in panel (a) of Fig. 2 as a common area between two different types of activity domains.

In an ideal recovery process, the intensity of physical reconstruction and socio-economic recovery activity remains at their

maximum level during the recovery time to maximise the functional recovery zone. However, this would be very challenging. As will be discussed in Sect. 3.2, resource allocation and concentration on the disaster area will not remain constant during the whole recovery period. External support would decline, and the intensity of reconstruction activities could decrease (Choi et al., 2019). However, it is important to note that the socioeconomic recovery process also requires physical infrastructure (Barakat and Zyck, 2011; Mitsova et al., 2019). Therefore, the intensity of related recovery activities may be raised with the

repair of some of the damaged facilities, while at the initial stage of the recovery, it might not be so high due to the damaged structures and infrastructures.

As shown in panel (b) of Fig. 2, by increasing the intensity of socio-economic recovery activities, the common zone (functional recovery) could be preserved or even increased during the recovery period. Furthermore, the intensity of socio-economic recovery activities could be increased without external support in a disaster-struck community (Alifa and Nugroho, 2019).

These cooperatively evolving activities enable people to take part in the restoration of their communities independently (Nigg, 1995; Talbot et al., 2020; Perce, 2007). For instance, as more enterprises of all sizes become involved in the economy, people will be more capable of actively participating in the economic recovery of their community (Freeman, 2004). Setting this

balance between physical reconstruction and socioeconomic recovery would be possible if the disaster area needs assessment (see Sect. 4.1) is considered.


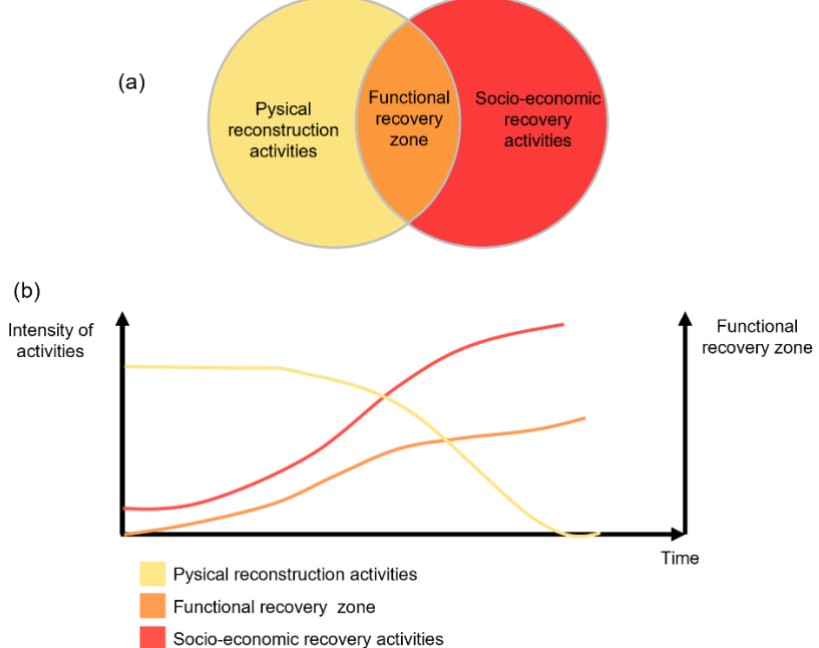

**Figure 2.** Panel (a): The relationship between physical (in yellow) and socio-economic (in red) activities in the recovery process, as well as their interaction zone (in orange), is called the functional recovery zone. Panel (b): Changes in the intensity of physical and socio-economic recovery activities over time, and the resulting changes in the functional recovery zone. The reader is referred to the web version of this
article for an interpretation of the references to colour in this figure.

Assessing system performance and the effectiveness of investments made in various sectors throughout the process of recovery following a disaster, as well as quantitatively evaluating recovery plans for upcoming events, has always been a challenge for decision-makers and researchers. Although many frameworks and metrics have been developed to evaluate recovery efforts following a disaster in various aspects, such as physical reconstruction (Charles et al., 2021; Labadie, 2008), economic
(Marshall and Schrank, 2014; Yamaguchi et al., 2016), and social recovery (Bahmani and Zhang, 2021; Dwyer and Horney, 2014), measuring the recoverability of various communities by evaluating their recovery plans, pre-disaster investments and resilience has received less attention (Berke et al., 2014). Social stratification in urban systems can result in differential rates of recovery among diverse social classes, races, ages, genders, and family statuses (Fussell, 2015; Cutter et al., 2006). Moreover, recovery measurement on various scales (e.g., individuals, groups, communities, cities, etc.) would necessitate the
use of ad-hoc indicators and methodologies (Chang, 2010).

## 3.2 Recovery in the DRM cycle

The DRM cycle is widely recognized in the global DRM community as a framework for managing various types of disasters, both natural and anthropogenic (Coetzee and Van, 2012). It is characterized by separate and sequential phases with varying durations and actions. While phases' number and their naming vary in the literature, the following main ones (before, during, and after the event) can be identified: preparedness and mitigation, response, and recovery. However, the current understanding of recovery recognizes it as an ongoing, long-term process that can start simultaneously with the response phase, and the developmental recovery activities are extended alongside the mitigation phase, leading to the overlap of different phases in practice (Twigg, 2015). Moreover, the current DRM cycle falls short of effectively addressing the complexities of multi-(hazard-)risk scenarios. The DRM cycle, characterised by separate and sequential phases, does not adequately capture the dynamics and interaction of these multiple hazards, particularly those involving both sudden-onset (e.g., earthquakes, flash floods) and slow-onset hazards (e.g., pandemics, droughts, conflicts) (Terzi et al., 2022). Consequently, numerous authors have proposed alternative frameworks for DRM, challenging the current circular representation (Bosher et al., 2021; Staupe-Delgado, 2019; Terzi et al., 2022). Despite the considerations, the conventional DRM cycle continues to prevail as the predominant discourse within the realms of decision-makers, practitioners, and the academic community focused on disaster risk, and it is used as a reference for the current research.

The definition of the starting time of the recovery process represents a controversial issue in the literature and is sometimes impacted by the DRM cycle structure. According to Rotimi et al. (2009), recovery is the phase just after the initial response to a disaster, while for UNDP (Arenas et al., 2016) recovery activities begin immediately after a disaster, during the relief phase, and continue until full recovery is accomplished. More specifically, coordination and recovery planning are the two main activities to start immediately after a disaster and concurrently with the response phase (Arenas et al., 2016). Empirical data and previous experiences show a disparity in allotted resources, organisation, and contributions between the response and recovery phases (Ali et al., 2020). However, it should be noted that experience indicates that addressing the short-term requirements of affected populations during the response phase has an influence on meeting the needs of the population during long-term recovery, and addressing these two types of needs should be done in an integrated way (Garnett and Moore, 2010). As emergency requirements settle and the media spotlight fades, consideration is given to the long-term implications of loss. Nonetheless, the longer and more costly phase of disaster recovery rarely receives the same level of support compared to the response, even though it may influence a community's future well-being for years to come (Choi et al., 2019; Davis, 2007; Raju and Becker, 2013).

Transitioning from the response to the recovery phase has not been adequately addressed in the literature (Levine et al., 2007). The questions of how, when, and who will organise, finance, and manage this transition process remain unanswered (Ali et al., 2020; Muskat et al., 2015). Trying to answer this question, the UNDP (Arenas et al., 2016) strategy entails breaking down recovery into stages, named early, medium, and long-term recovery, respectively. Early recovery occurs during the transition period from emergency relief to long-term restoration. More specifically:

- Early recovery begins with quick interventions such as financial support for reconstruction or food.
- Medium-term interventions focus on rebuilding shelters, infrastructure, and livelihoods.
- Long-term recovery concerns strengthening government capacity and lowering the risk of future disasters.

Analogously, with the goal of evaluating the success of the recovery initiatives, Bahmani and Zhang (2021) separated the recovery process into three distinct time domains: short-term, medium-term, and long-term. For each time domain, they identified a series of critical success factors. Short-term critical success factors refer to the ability to answer people's emergency requirements, which should primarily be addressed during the response phase (Bahmani & Zhang, 2021). As a result, the short-term recovery phase as defined by Bahmani and Zhang (2021) coincides with the response phase.

The practice of dividing recovery into sub-phases is not new. Kates and Pijawka (1977) proposed a four-phased sequential description for recovery, which included an 'emergency', 'restoration', 'replacement and reconstruction', and a 'commemorative, betterment, and developmental reconstruction' period. Even though their model has been referenced and used in several publications (Kates et al., 2006; Platt and So, 2017), it has some flaws (Sobhaninia and Buckman, 2022; Rubin, 2009). Sobhaninia & Buckman, (2022) criticised the Kates & Pijawka model for not considering pre-disaster statutory and different types of disasters. As an alternative, they presented a reconfigured model based on Kates and Pijawka's approach, emphasising anticipation, equity, and resilience. They incorporated in their model phases' overlaps and varying durations of sub-phases (emergency, restoration, replacement, and reconstruction) that are not always consecutive for all communities, as well as pre-disaster vulnerability and equity and their impact on recovery. Furthermore, by including the resiliency notion in their model, they considered the recovery effect on pre-disaster conditions, making it a cyclic procedure (Sobhaninia & Buckman, 2022).

The recovery in L'Aquila following the earthquake in 2009 is a good illustration of how the boundaries between the various stages of recovery are not always clear and can be fuzzy. In April 2009, L'Aquila, Italy was hit by a 6.3 MW earthquake with a hypocentre depth of 10 km. Reconstruction efforts led to the construction of 5722 housing units in 19 new settlements distributed around the city within one year of the earthquake (Contreras et al., 2013). The newly built housing included features like seismic isolation and solar cells, typically associated with the development phase of recovery. However, by 2014, damaged houses could still be found in the city centre, indicating the presence of the early recovery phase. The simultaneous presence of early recovery and development recovery actions highlights the fuzziness of boundaries between different phases of post-disaster recovery (Contreras, 2016; Contreras et al., 2014).

Figure 3 depicts how the functionality of an urban system affected by a disaster may change over time. In this study, the urban system functionality is described as the effective and interdependent operation of infrastructure, services, and socio-economic activities within a city to meet the needs of its population and safeguard them against potential hazards. The phases of the DRM cycle are depicted in the background, with the system behaviour in each phase is indicated by a black line. According to Zhang et al. (2021), the system can have a dynamic behaviour in the pre-disaster period (segment A-B) depending on the decisions made by stakeholders, and the general trend could be ascending or descending. In the hypothetical example provided in Fig. 3, before the disaster, stakeholders are attempting to improve the system's functionality. Immediately after the disaster,

there would be a reduction in functionality (segment B-C), and then a consequent increase due to the actions that may be carried out during the response phase (segment C-D). A slight decline in functionality is expected during the transition period

due to the removal of external support (Choi et al., 2019; Davis, 2007; Raju and Becker, 2013), followed by an upward trend in functionality during the recovery phase. A noticeable point is that pre-disaster decisions and actions influence **α** and **β**, which are the slope of the decreasing and increasing functionality during the disaster and recovery respectively (Sobhaninia and Buckman, 2022).

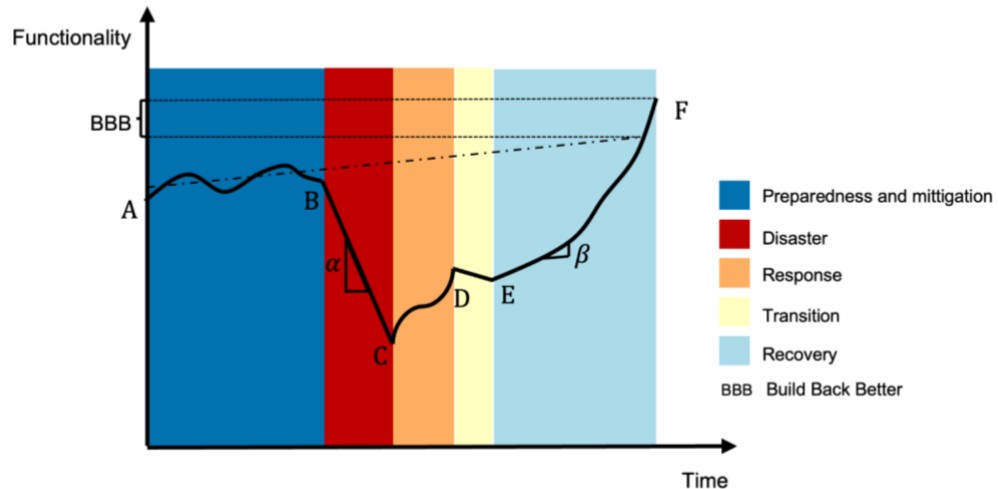

**Figure 3.** Urban system state behaviour in terms of functionality in different DRM phases includes: the dynamic fluctuation of functionality due to the lack of maintenance or mitigation activities that are conducted with various intensities during preparedness time according to the stakeholders' decision (segment A-B); the abrupt decline in functionality as a consequence of a disaster occurrence with a given slope **α** which depends on system preparedness and robustness in pre-disaster (segment B-C); increase in the urban system's functionality in a short time and with a noticeable slope due to the external contribution and sources during the response phase (segment C-D); transition phase

between response and recovery and a slight decrease in functionality of the urban system due to the lifting of external support and public attention fading (segment D-E); recovery phase and increase in system's functionality with a given average slope **β** which depends on pre-disaster and post-disaster planning and activities (E-F); final urban system's functionality level (point F), which could be higher than its initial value in pre-disaster time (point B).

The dynamic represented in Fig. 3 is commonly used in the literature to describe the urban system's functionality (Choi et al.,

2019; Fang et al., 2016; Ghorbani-Renani et al., 2020; Zhang et al., 2021). Nevertheless, such a representation does not consider the nonlinear increase of functionality during recovery. As previously stated, the recovery actions are not always carried out in a sequential manner, and therefore the functionality is not necessarily increasing as a continuous line, as is shown in Fig. 3 (Rubin, 2009). This current issue underlies the need for functionality models able to capture non-sequentially behaviour in the recovery process.

 **3.3    The final goal of the recovery process**

Researchers did not agree on where the recovery process should lead and conclude (Ganapati and Mukherji, 2014; Ismail et al., 2014; Sadiqi et al., 2017). Defining the recovery procedure's goal depends on many factors and doing so without considering the system's condition in terms of functionality in the different phases of the DRM cycle can lead to simplistic or useless results. The least that may be expected from the recovery is a return to pre-disaster conditions (Quarantelli, 1999; Su

and Le Dé, 2020; Whyte, 1979). However, there is no certainty that this will be achieved. Unawareness of different options in defining objectives for recovery procedures and the lack of participation of all stakeholders in the recovery decision-making stages may result in the community's inability to return to pre-disaster conditions, or worse yet, increase their exposure and vulnerability (Smith and Wenger, 2007). Poor reconstruction as a result of focusing solely on a quick recovery, job losses, a reduction in affordable housing stocks, and the inability to assist the disadvantaged classes of the community in their recovery

are just some of the possible repercussions of the bad recovery that results in an urban system that is even more vulnerable than it was before the disaster (Bolin and Bolton, 1986; Peacock et al., 1997; Vale and Campanella, 2005).

As illustrated in Fig. 3, the endpoint of the recovery process (point F) can even be higher than its initial value in pre-disaster time, highlighting how the recovery can represent an opportunity for improvement in urban systems. The notion of disaster as an opportunity has progressively gained traction in different fields, such as technological, economic, and social.  Shaw et al.

(2003) have demonstrated that meaningful change is possible under the right circumstances after a disaster. Most of the time, the savvy communities that survive after a disaster have the chance and ambition to work toward improving the citizens' economic, and environmental conditions, and quality of life (Smith and Wenger, 2007). Instead of perceiving recovery as a return to the status quo, the viewpoint of disaster as an opportunity laid the groundwork for the rise of concepts like 'new normal' as the final stage of recovery, which may be viewed as an adaptive process that negotiates the conflicts between re-

establishment of pre-disaster systems and considerable transformation of those systems (Smith and Wenger, 2007; Tierney and Oliver-Smith, 2012).

In the field of economics, the recovery endpoint is defined as 'a-priori' as the condition that would have been attained if the disaster had never happened. This method highlights the impact of ongoing trends outside of the disaster on indicators like unemployment rates and house prices (Cheng et al., 2015). For example, considering the GDP per capita as an index, Chhibber

and Laajaj (2007) propose certain long-term scenarios concerning the index's behaviour. Instead of using the exact amount of GDP prior to the disaster's occurrence, they used other criteria. They used the pre-disaster GDP growth rate to linearly extrapolate the GDP the community would have reached at the recovery endpoint if the disaster had not occurred. They claim that even though a disaster will inevitably result in a decline in GDP per person due to the extensive damage to the capital stock, it is still possible to use the disaster as an opportunity to achieve a higher GDP per capita at the end of recovery.

The concept of 'new normal', as an improved status quo of economic and social systems, is strictly connected to the concept of BBB. In a report from 2006 titled "Key Propositions for Building Back Better", former US President William Clinton first introduced the BBB approach to disaster recovery (Clinton, 2006). Since then, BBB (see definition provided in Sect. 3.1) has

become the catchphrase of post-disaster reconstruction programs, to the point where the second half of Priority 4 of the Sendai Framework for Disaster Risk Reduction 2015–2030, was designated to BBB. The BBB concept has undergone numerous interpretations depending on stakeholders' perspectives (Fernandez and Ahmed, 2019; Kennedy et al., 2008). The adjective 'better' has been interpreted as 'safer', 'greener', 'more environmentally friendly', 'more aesthetically appealing', 'more oriented toward livelihoods', 'more resistant to natural hazards', 'faster', 'stronger', and 'more equitable', among others (Hinzpeter and Sandholz, 2018; Kennedy et al., 2008; Kim and Olshansky, 2014). Moreover, measuring quantitatively the success of the BBB is another controversial issue (Thomalla et al., 2018; Fernandez and Ahmed, 2019). Therefore, Tatham and Houghton (2011) question still needs to be answered: "Who decides what 'better' actually means?".

BBB can be successfully exploited from a multi-risk disaster perspective. By incorporating BBB into resilient post disaster recovery efforts, not only the impacts of the disaster that imposed damage are addressed, but also the potential future hazards that the urban area may face are considered. This comprehensive approach reduces the vulnerability of the urban system to multiple risks and can potentially minimize damage in the event of another hazard (De Angeli et al., 2022). An example of a resilient recovery measure is the implementation of changes to building standards (Fernandez and Ahmed, 2019; Kennedy et al., 2008) In Fig. 3, BBB at the end of recovery has been determined using the economists' notion as mentioned above, considering the pre-disaster trend in terms of functionality growth. While this idea may appear ambitious for a progressive urban system in terms of functionality, considering the advantage of using the external resources that would enter the system during the recovery, reaching such a position would be feasible.

## 3.4    Issue 1 key points

The main key points about the recovery and its role in the risk management cycle are summarised hereafter:

- The concept and definition of recovery have evolved over time passing from merely focusing on physical reconstruction towards inclusion of socio-economic aspects of recovery in the definition.
- The distinction between response and recovery phases is still a controversial issue and has been the motivation for defining the transition phase between them.
- Recovery planners typically view recovery as a sequential and predefined process, divided into subphases. However, real experiences challenge this perspective, as the recovery process is often more complex and unpredictable.
- Determining the goal of recovery, such as achieving resilience through a BBB approach to prevent future disasters, adds complexity to recovery planning, especially in multi-risk environments.

# 4    Issue 2: Recovery requirements in the current world setting

## 4.1    The importance of community needs in recovery decisions

Recovery is a long-term process that might take years or decades to complete (Dunford and Li, 2011; Olshansky, 2006), and ignoring socio-economic aspects of the involved urban systems throughout this time will impede comprehensive community development during and after recovery (Oliver-Smith, 1990). Some early studies (Rubin and Barbee, 1985) on recovery equated socio-economic recovery with physical reconstruction. Socio-economic recovery necessitates the restoration of some physical structures to offer spatial conditions for shaping social and economic linkage and communication. Restored socio-economic connections, on the other hand, can influence and improve the physical recovery process. As a result, while these two realms have mutual effects, they are clearly distinct and are characterised by different requirements and actions in the recovery process (Tierney and Oliver-Smith, 2012). Paying attention to people's needs in recovery decisions is essential for moving toward a holistic approach that addresses socio-economic and physical recovery in constructive connection with each other. However, prevalent recovery practices have shown that mostly governmental assistance programs draw more attention to physical recovery planning than systematic identification of community needs (Comerio, 2014). During an electoral cycle, politicians often display a tendency to prioritize physical reconstruction efforts over socio-economic recovery. This preference entails a stronger focus on rebuilding infrastructure and physical structures, driven by political motivations to demonstrate tangible progress and garner public support (Masiero and Santarossa, 2021).

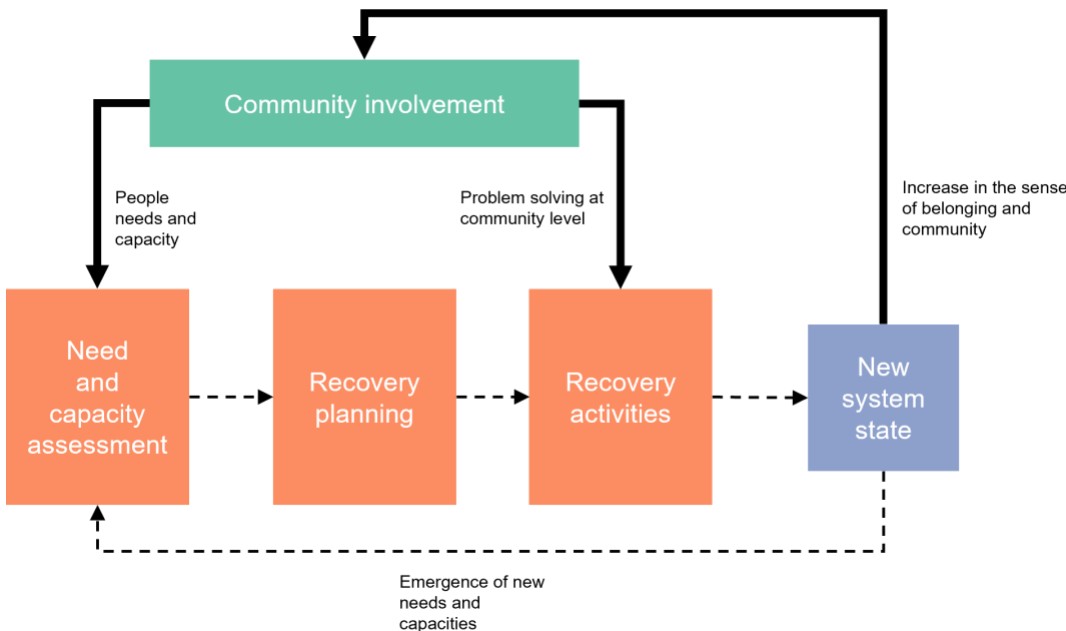

**Figure 4.** Role of community involvement in need and capacity assessments, recovery planning, and recovery activities. The need and capacity assessment represents a key element in forming and preserving the dynamic process of recovery and the incremental participation of the community in the process.

Comprehension of community needs integrates visions into recovery plans toward building long-term resilience in disaster-affected communities and making them capable of facing upcoming events (Chandrasekhar, 2012; Iuchi et al., 2015). A common challenge that arises in the recovery process following a disaster is balancing the need to enhance resilience against future disasters with the survivors' desire to quickly rebuild their houses and livelihoods (Esteban et al., 2015). This dilemma becomes even more complex in the case of multiple hazards affecting the disaster-prone area. It becomes challenging to persuade stakeholders to allocate resources towards increasing resilience against hazards other than the one that recently caused damage. For instance, consider a community that has been recently affected by an earthquake, resulting in the loss of homes and livelihoods. In such a situation, it may not seem reasonable for the community to invest in reconstructing dikes for flood prevention. However, if a different hazard, such as a flood, occurs subsequently, it can have a noticeable impact on the community's recovery. Therefore, it is crucial to implement a co-production knowledge process that incorporates both local knowledge and scientific knowledge in order to ensure comprehensive consideration of all dimensions of resilience during the recovery process (Šakić Trogrlić et al., 2021).

Community needs, and capacity assessment is also required for recognizing and delivering solutions to problems that can and should be tackled at the community level (Gaillard and Mercer, 2013; Horney et al., 2016; Bolin and Stanford, 1998). As a result, as shown in Fig. 4, community involvement might occur in two types of interventions during the recovery process:

    i.    In needs and capacity assessment, to properly integrate community needs and planning recovery accordingly.

    ii.    In recovery activities, using the existing capacity to tackle the problem at the community level and efficiently employ local resources.

Furthermore, community involvement can be intensified as much as the disaster recovery process is able to address their needs (Iuchi, 2014; Sovacool, 2017). As illustrated by the arrow that connects the 'new system state' to the 'Community Involvement' in Fig. 4, the more the recovery activities are directed toward meeting the needs of the community, the more the population will find the recovery process effective and will be encouraged to participate because they find themselves as an important part of decision-making process (Mutch, 2014). As a result, since the community's resources and capacity would be mobilized in an efficient and influential manner toward progressive recovery, it could be noted that while the need assessment initially appears to slow down the recovery process by delaying tangible actions, it ultimately speeds it up in the long term (Chandrasekhar et al., 2014).

465

Table 2. Post disaster need assessment approaches conducted in six different publications (the complete table is available in the supplementary material)

| Author(s) | Assessment target | Method | Input data / considered factors | Hazard | Main priorities amongst the needs | Involved stakeholders |
|---|---|---|---|---|---|---|
| He (2018) | Community needs | Semi-structured interviews and focus groups | - Disaster impacts<br>- Disaster public perception<br>- Local capacity<br>- Expected external aids | Earthquake | - Productive living and development issues<br>- Permanent ownership of land<br>- Gain of employability for overseas job-hunting | Earthquake-resettled households and community leaders |
| He (2019) | Community needs | Semi-structured interviews and focus groups | - Disaster public perception<br>- Expected external aids | Earthquake | - Safe land for reconstruction<br>- Constructive conversation with the government<br>- Permanent housing<br>- Land grants | Earthquake-resettled households and community leaders |
| He et al. (2019) | Evaluation of the current living conditions (compared to the pre-disaster) | - Semi-structured interviews and focus groups<br>- Documentary and field investigations | - Physical living conditions<br>- Off-farm job availability<br>- Living expenses<br>- Place attachment<br>- Expected external aids | Earthquake | Non-agricultural employment | Households |
| Kurosaki (2017) | Assessment of the level of household recovery | Panel survey | - Preliminary disaster damage assessment<br>- External aids received<br>- Changes in the productive assets | Flood | Productive assets functionality | Households |
| Deen (2015) | - Vulnerable households and undermined informal coping strategies<br>- Government's capacity | - Review and analysis of the existing literature and documents<br>- Interviews | - Reducing vulnerability<br>- Sustainable development goals | Flood | - Public health services<br>- Sustainable shelter<br>- Restoration of on and off-farm<br>- Incomes (agriculture and livestock)<br>- Public administration<br>- Infrastructure<br>- Education services | Key government officials |
| Honjo (2011) | Life recovery assessment | Workshops | Citizens' happiness | Earthquake | Permanent housing | Citizens |

Table 2 reports a series of post-disaster need assessment approaches presented by different authors in the literature. It tried to find studies that followed, established, or reviewed applicable methodologies for needs identification and assessment and applied it to at least one real-world case study. The papers that only discuss recovery or early recovery are chosen for this collection. If a paper just addresses institutional or other forms of shortages, it is not considered because the chosen papers tackle needs that require physical assets to be addressed. Moreover, the articles included deal with prioritizing and comparing community needs rather than exclusively locating resources to address a particular need in a specific area. While a complete table is provided in the supplemental material, Table 2 attempts to focus on selection of studies mainly including stakeholder participation. Malilay et al. (1996) used quantitative cluster-sampling to conduct a rapid needs assessment in the aftermath of disasters, focusing on population-based data including remaining population and the number of people with specific health needs. In assessing the needs, their model considered disaster damage, the number and types of specific health needs, and the number of housing units. To assess the post disaster health needs consequent to the 1999 Chi-Chi earthquake in Taiwan, Chen et al. (2016) adapted this cluster-sampling technique to the community needs assessment and morbidity and mortality surveillance, which mainly considered population data and infrastructure functionality after damage in the quake areas.

The United Nations Development Group (UNDG), the World Bank (WB), and the European Union (EU) worked on the development of a Post-Disaster Needs Assessment (PDNA) tool ( GFDRR, 2013). This tool provides an objective, thorough, and government-led assessment of post-disaster damages, losses, and recovery needs, setting the path for a consolidated recovery framework. The PDNA Guide (GFDRR, 2013) gives step-by-step guidance on planning for and implementing a PDNA, drawing on and incorporating several assessments and planning methodologies such as the Damage and Loss Assessment (DaLA) and the Human Recovery Needs Assessment (HRNA). The tool has been implemented in several case studies (Hinzpeter and Sandholz, 2018; World Bank Group et al., 2016; World Bank Group, 2017).

Considering the different studies reported in Table 2, it is possible to see that most of the approaches and methodologies for the recovery needs assessment are qualitative, relying on interviews and document surveys, and many of them focus on earthquake disasters. Most of the studies emphasised employment, jobs, and housing as primary concerns. Stakeholders and researchers all around the world have been debating whether housing or employment should take precedence over other recovery priorities, and they have yet to agree on which should be prioritised (Dunford and Li, 2011; He, 2019). Prioritisation should not lead to decision-makers ignoring other essential concerns in recovery and directing all resources to meet only the most demanding needs. For example, if a community identifies housing as their primary need, focusing on new housing construction and allocating most recovery resources to this sector may lead to a misunderstanding of recovery progress and the withdrawal of government aid and intervention before true recovery in all sectors, including the economy, occurs (Lyons, 2009).

Shortening reconstruction projects by directing all financial resources toward physical reconstruction may result in growing construction-related businesses and temporary jobs for locals. Consequently, it would appear to decision-makers that the economic recovery is advancing. However, after the reconstruction is complete, neglecting sustainable economic growth would

present a new challenge that would be more difficult to deal with due to the allocation of the majority of financial resources earlier to physical reconstruction (Dunford and Li, 2011).

Furthermore, the needs are interconnected, and treating one may result in the solution of another need, or the sort of answer offered to a need may lead to the formation of new demands (He, 2019; Zhang, 2016). Moreover, as the perception and needs of the affected people change in the aftermath of a disaster, the solutions offered by the decision-making for recovery should

change as well (Chandrasekhar, 2012; Kurosaki, 2017). For instance, relief and short-term recovery efforts are critical and urgent, with the goal of shortening the time it takes for people to reclaim a safe home and secure livelihood. However, redevelopment policies should be carefully developed based on comprehensive, site-based risk and vulnerability assessments, as well as ongoing consultations with all stakeholders (Downing, 2002; Ingram et al., 2006). All this implies that determining recovery needs is a dynamic process that should be continued throughout the recovery period (Downing, 2002; Ingram et al.,

2006). As illustrated in Fig. 4, one of the major components of the recovery plan should be the needs and capacity assessment of the community, and after each recovery cycle, and reaching a new system state in terms of functionality, a new need assessment should be undertaken to identify newly emerging needs. The L'Aquila C.A.S.E. reconstruction project is an excellent example of the mismatch between people's needs and priorities during the early-recovery phase and in the long term. Specifically, the project was initially conceived as 'temporary housing' but later resulted in a permanent housing solution that

failed to meet the residents' long-term needs (Alexander, 2010). According on the population's needs changing during various phases of recovery, some urban subsystems may need to operate differently. For instance, during long-term recovery, the health system should place more of its effort on treating chronic diseases and mental illnesses rather than providing emergency care which is the focus area of these systems during response and early recovery phase (Runkle et al., 2012).

In addition, post-disaster transformational changes (Blackburn, 2018) should be constructive or would cause dissatisfaction

among the target group. This is the case of post-disaster relocation programs, in which people are typically resettled in another location unwillingly without regard to their pre-disaster socioeconomic status (Shanmugaratnam, 2005), which mainly leads to a sluggish recovery process (Charny and Martin, 2005). People are unwilling to alter their living habits to reduce their exposure to a natural hazard if it increases their vulnerability to other threats like economic insecurity. Therefore, exposure reduction should be one of the top priorities for choosing the location of the resettlement but not the only one (Davidson et al.,

2007; Degg and Chester, 2005; Shanmugaratnam, 2005). In the case of L'Aquila post-earthquake recovery, population relocation (Mannella et al., 2017) without consideration of issues such as sufficient urban facilities (Forino, 2015), spatial connectivity (Contreras et al., 2013), social fragmentation, lack of functional living, and questionable ecological values (Alexander, 2010a; Özerdem and Rufini, 2013) resulted in an incomplete and slow recovery process and eventually a not resilient city (Contreras et al., 2017). Other flaws in L'Aquila recovery include focusing solely on meeting the population's

quantitative needs while ignoring the quality of life of the afflicted population (Alexander, 2013) in housing and deferring the reconstruction of the historical areas while ignoring their role in the city and citizens' identity (Contreras et al., 2014). Moreover, pre-disaster community involvement in recovery planning and social organization plays a pivotal role in enhancing the community's capability to leverage its capacities, particularly its social capacity, to effectively address challenges in the

aftermath of a disaster and actively engage in the recovery process. Social organization through the involvement of different

stakeholders in pre-disaster planning significantly contributes to the successful implementation of recovery plans and facilitates the realization of post-disaster recovery efforts (Delilah Roque et al., 2020).

An illustrative example of the significance of social organizations can be observed in the aftermath of Hurricane María in Puerto Rico. During this time, social organizations played a crucial role in coordinating and addressing the comprehensive needs of the community. Despite the island-wide power outage, these organizations became vital hubs where community

members could come together to organize, support one another, and address the challenges faced. Their pre-disaster planning efforts and established networks allowed for effective coping mechanisms and resource allocation, even in the absence of immediate access to personal finances held in banks (Delilah Roque et al., 2020).

## 4.2    Multi-hazard risk recovery requirements

This section aims to explore additional recovery requirements posed by multi-(hazard-)risk conditions. The requirements have been identified by analysing different strategies available in the multi-(hazard-)risk recovery planning literature. The analysis encompasses an evaluation of the strengths and limitations of the different available approaches. The discussion is supported by real-world examples.

Table 3 includes a selection of studies that illustrate the key approaches for recovery planning in a multi-hazard risk context.

Only four studies were included in this table as representative of the main multi-(hazard-)risk approaches available in the literature. The complete table of the analysed approaches is available in the supplementary material. It should be noted that only the papers that addressed the recovery in a multi-(hazard-)risk environment were considered. Hence, even though there are many more studies that account for multi-(hazard-)risk assessment or multi-hazard risk resilience evaluation, they are not included in the table because they do not address recovery specifically or because they do not clearly discuss the application

of their methods in recovery planning.

Some researchers tackled multi-risk issue in recovery planning through prioritising one of the hazards over the others in order to implement DRR measures based on risk comparisons (Der Sarkissian et al., 2020, in Table 3). Consequently, a significant portion of resources for risk reduction measures is allocated to reduce the risk of the so-called 'dominant hazard' (Nakanishi and Black, 2018). However, this perspective shows some inefficiencies, particularly during the recovery phase when the overall

system is more vulnerable due to the impact of the dominant hazard.


Table 3. A selection of four publications presenting multi-(hazard-)risk approaches in recovery or resiliency analysis (the complete table is available in the supplementary material)

| Author(s) | Considered hazards | Included territorial element(s) | Spatial scale | Aim of the research | Multi-hazard approach |
|---|---|---|---|---|---|
| Der Sarkissian et al. (2020) | - Earthquakes<br>- Wildfires<br>- Floods<br>- Windstorms<br>- Landslides | Road network | Country | Assess road network resilience to natural hazards. | Multiple hazards are considered in a separate way (i.e., without considering any interaction) and compared. |
| Gentile et al. (2019) | - Earthquakes<br>- Tsunami | Schools | Urban area | Prioritisation for DRR measures | The considered hazards are combined in a multi-hazard index |
| Cheng et al. (2021) | - Earthquake<br>- Heavy wind<br>- Cyber-attack | IEEE 9-bus as an abstract and approximation of country power grid system | Country | Quantifying the resilience of engineered systems under random multi-hazard by employing availability as an effective performance indicator and assessing the system's availability across time, based on which resilience is quantified w.r.t. system robustness and recovery ability. | Interdependent hazards are considered, and their interdependences characterized by the copula, joint distribution, and Markov models. Cascading failures have taken into account. |
| Argyroudis et al. (2020) | - Flood<br>- Earthquake | Multi-span highway bridge | Structure | Life-cycle assessment of the resilience of infrastructure, by considering all the possible events that may affect the system during its design lifetime. | Physical vulnerability surface that includes both hazards and accumulated damage due to the occurrence of the second hazard before complete recovery from the first one. |


The main challenge with prioritizing dominant hazards in DRM is that the ranking of hazards in terms of their potential impact might change after the occurrence of the 'dominant hazard' and during the recovery process. This is due to changes in the vulnerability of the community (De Angeli et al., 2022). Therefore, relying solely on the dominant hazard perspective may lead to overlooking other significant hazards and their potential consequences, putting the recovery efforts and the affected 580 population at higher risk. In Japan, disaster preparedness efforts primarily focus on earthquake-related risks due to their significant threat to local communities (Nakanishi and Black, 2018). Government initiatives and university programs

emphasize disaster risk reduction measures concerning earthquakes (Nakanishi and Black, 2018). However, there is often not enough focus on land-use planning for flood mitigation and evacuation modelling or in general flood disaster risk management (Nakanishi and Black, 2018). This oversight becomes particularly problematic when considering the occurrence of floods following earthquakes and during the recovery period. The occurrence of earthquakes can destabilize flood defence structures and increase the likelihood of subsequent floods, catching the population off guard and unprepared (Liu et al., 2009).

Another approach to multi-(hazard-)risk recovery (e.g., Gentile et al., (2019)and Sevieri et al., (2020)) is based on the integration of two risk indices into a single multi-hazard index for decision-making and prioritising among various DRR measures for a building portfolio. Nevertheless, these DRR measures are not fully investigated from a multi-(hazard-)risk perspective. While a DRR intervention can help in decreasing the risk of a single hazard type, it can also have unwanted effects on other hazard risk typologies, such as an increase in the vulnerability (Crosti et al., 2011; Li et al., 2012). These effects are referred to as "asynergies" of DRR measures (de Ruiter et al., 2021). However, in the multi hazard index-based approaches asynergies and accumulated damage caused by the potential consecutiveness of multiple hazards are overlooked.

In their analysis, Cheng et al., (2021) considered a simplified representation of a real-world power grid to assess its multi-hazard recoverability. Although they considered the likelihood of the second hazard occurring during the period of recovery from first one, they neglected to account for the interaction between these two types of hazards and the consequently greater impact on the system when compared to the occurrence of single multiple hazards. Indeed, in the case of consecutive disasters, the exposed elements could remain in an unrecovered state due to the limited time interval between the two consecutive hazards, and as a result, the second hazard's occurrence would cause greater impacts than it would have generated without previous damage (De Angeli et al., 2022; de Ruiter et al., 2020; Gill and Malamud, 2016; He et al., 2018).

Argyroudis et al., (2020) proposed a more comprehensive approach compared to other works, by considering the interaction between multiple hazards in the recovery process. They analysed the recovery dynamics and accumulated damage caused by multiple risks, focusing specifically on a single infrastructure (i.e., a bridge), to assess its resilience. Nevertheless, they examined this infrastructure in isolation, without considering interdependencies with other infrastructures. Ignoring such a systemic perspective could lead to neglect important factors, such as the transportation of reconstruction materials and human resources, resulting in unrealistic recovery planning for the considered infrastructure (Zamanifar and Hartmann, 2020; Der Sarkissian et al., 2022).

In the context of multi-hazard recovery, it is crucial to consider not only the spatial and temporal overlap of hazards, but also their collective impact on shared resources utilized for the recovery process. The economic aspects of recovery planning have been addressed only from a single-hazard perspective but could play an even greater role in multi-(hazard-)risk recovery because of the additional challenges posed by the multi-risk environment. One important aspect to address is the influence of these hazards on financial resources and the overall cost of recovery. After a disaster, governments require funds at various stages of the recovery process (Ghesquiere and Mahul, 2010). While the physical reconstruction stage demands the greatest amount of funding, significant resources are also needed for relief and early recovery activities such as debris removal and temporary shelter provision (Ghesquiere and Mahul, 2010). This allocation of funds for immediate needs can limit the available

resources for long-term reconstruction efforts. The occurrence of multiple hazards in a given region not only imposes substantial costs on the government but also affects insurance companies providing catastrophe insurance. In Japan, for instance, the combined losses from the five major events in 2018 did not surpass the costs of the most devastating earthquakes, floods, and typhoons experienced in recent years. However, due to the severity of the typhoon season and associated rainfall,

2018 still ranked as one of the costliest years in terms of losses incurred (Fujimura, 2019). This highlights the importance of considering the cumulative impact of losses across multiple perils. While individual large-scale events may be manageable within a reinsurance program, the accumulation of losses from various hazards poses a significant challenge if a holistic and aggregate perspective is not adopted in risk management practices (Fujimura, 2019). As a result, the development of a comprehensive tool that can account for asynergies, cumulative damages, recovery dynamics, and any other interaction at

various levels (hazard, vulnerability, exposure, or DRR measure) is required to assist decision-makers in long-term recovery planning and DRR measures implementation to increase the resilience of the urban system (de Ruiter et al., 2020; Durham, 2003).

A comprehensive representation that shows how multiple hazards can influence the functionality of an urban system in the different distinct temporal domains with respect to the disaster occurrence time (pre-disaster, during disaster, and post-disaster)

is presented in Fig. 5. During the disaster time (represented by the red time frames in the "time d" timeline of Fig. 5), most of the response actions facing various hazards are carried out. If certain measures, such as propping damaged buildings, which primarily overlap with early recovery actions, are not correctly completed, or ignored entirely in this time frame, the system's functionality will continue to decline even after the disaster has occurred, as illustrated in panels A, B, and C in Fig. 5. Additionally, events of any size, no matter how severe, that occur after a destructive event may result in the system's

functionality being reduced because the system will be more vulnerable than it was prior to the big event, due to the damages that have been imposed by the first big event. This pattern may be seen in the panel C of Fig. 5, where the effects of shocks and landslides following an earthquake are depicted.

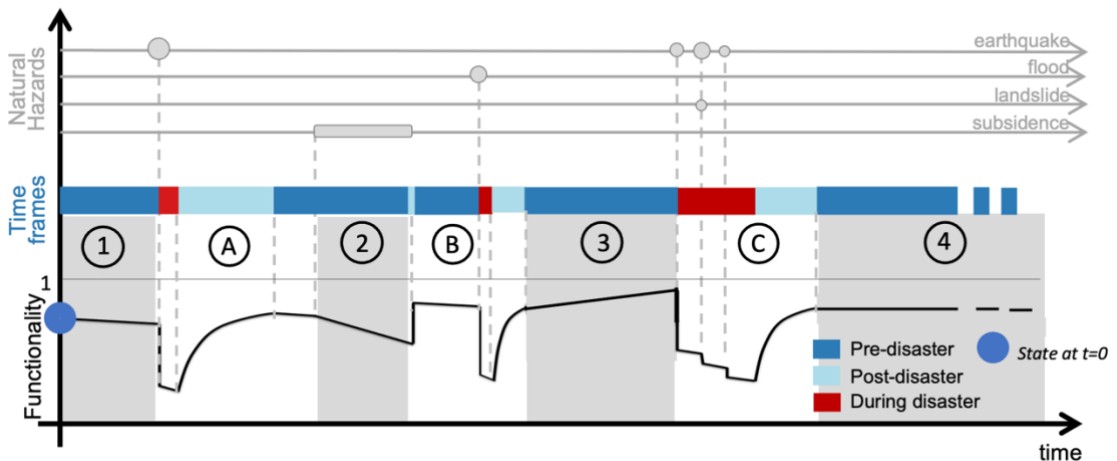

**Figure 5.** Possible evolution of the urban system functionality in a multi-risk environment divided into three time frames (pre-disaster, during a disaster, and post-disaster) , considering different types of hazards (rapid onset: earthquake, flood, and landslide; slow onset : subsidence) consecutively with various time intervals between them and different trends in pre-disaster ( ascending n. 3, descending n. 1 and 2, and steady n. 4) as a result of mitigation, lack of maintenance, and slow-onset hazards and their interaction with each other. Distinct variations in recovery rates are discernible across the white panels. Panel A illustrates the recovery process following a single rapid onset hazard. In Panel B, recovery ensues after a slow onset hazard, disregarding other potential hazards, resulting in a system vulnerable to other hazards such as floods. Nevertheless, the subsequent recovery rate, surpasses the rate of the recovery after the initial rapid onset hazard, namely the earthquake. Panel C highlights a segmented decline in system functionality, attributed to inadequate recoverability in the face of consecutive events happening within short intervals.

Furthermore, the pre-disaster time frame has a great effect on system functionality and plays a crucial role in the system state also during the other time frames. By increasing the functionality in pre-disaster, it is possible to reduce the loss caused by an upcoming disaster and prevent the system from reaching a very low level of functionality, from which it would be hard to recover. To investigate the role of the pre-disaster system's functionality, four different types of behaviour are identified in Fig. 5 by the grey panels numbered from 1 to 4. In panels 1 and 2, a gradual descending trend can be observed due to the lack of maintenance or the presence of slow-onset hazards (e.g., subsidence or coastal erosion). In time slot 3 the effect of mitigation and activities to increase the system's robustness has been depicted as having an ascending trend in the system's functionality. Time slot number 4 shows steady behaviour of the urban system's functionality because of the compensatory effect of the above-mentioned activities.

### 4.3    Issue 2 key points

The main key points about the recovery requirements in the current world setting are summarised hereafter:

- The post-disaster needs and priorities of the population, which should be at the core of recovery planning, vary across different social and geographic contexts.
- The needs of the population change over time in the aftermath of a disaster.
- Community involvement might occur both in the needs and capacity assessment phase, to tackle the problem at the community level, and efficiently employ local resources.
- Current approaches to multi-risk recovery planning only partially address the complexity of such topic, since: (i) they focus on hazard prioritization ignoring vulnerability dynamics exacerbated by consecutive disasters; (ii) they do not take into account potential asynergies between DRR measures, as well all damage accumulations; (iii) they do not evaluate systematically the interdependencies amongst different components of the urban system.

## 5    Issue 3: Decision making models and methods for investment in physical elements to improve recovery in urban areas

One of the main concerns of the decision-makers has been how to take cost- and time-effective resilience measures considering the resource restrictions (Ghannad et al., 2020). Making sure that the urban system runs to its full potential and that none of

675 the stakeholders or system components are malfunctioning because of the absence of necessary services that the other system components should be providing is a further concern that can aid in the efficient use of resources (Almoghathawi and Barker, 2019). Decision-making for investments to improve the recovery of widely diversified and broadening urban systems is especially challenging, which is why researchers have developed optimisation models that can help decision-makers in directing their resources (Zamanifar and Hartmann, 2020). Although there are many other types of optimization models that support allocating investments to various sectors to boost recovery, or descriptive frameworks for recovery planning that focused on different aspects of the system (Bozza et al., 2015; Choi et al., 2019), in this section, optimization models refer 680 only to models that distribute resources to the system's physical elements.

Optimization models for recovery planning available in the scientific literature can be divided into three classes (pre-disaster, post-disaster, and pre & post-disaster models) as shown in Table 4, according to the phase where the planned actions should be applied. As it can be seen in Table 4, the models were pursuing three different objectives (cost minimization 'C', recovery time reduction 'T', and serviceability maximisation 'S'), with some of them seeking two of these goals. It should be noted that 685 for assigning the objectives to the models that have been addressed, constraints that were considered by the authors in developing their optimisation model have not been regarded but the main goals of the optimization model were considered as the objective. The territorial elements with which the models were concerned were also factored in classifying them. As a result, lifeline infrastructure networks (electricity, water, gas, telecommunications, and roads) are the most frequently applied territorial elements in optimization models. Given that the value of lifeline infrastructures in the recovery phase is primarily 690 determined by their ability to provide service to residents and that they are the most covered territorial elements in the investigated models, serviceability is the most common objective in evaluated research. Although there have been many previous studies on some of the territorial elements, such as road networks, for which Zamanifar and Hartmann (2020) published a specific review on the recovery optimization model, in this study, it was attempted to select papers to cover a diversity of territorial elements.

Interdependencies play a crucial role in the functioning of critical structures and infrastructural systems (Ouyang, 2014). Even though different studies (e.g., Dudenhoeffer et al., 2006; Rinaldi et al., 2001) proposed various types of interdependencies in urban system modelling, such as physical, cyber, geographical, spatial, mutual, collocated, etc., we only considered generically the presence of any type of interdependency between more than two sub-systems in our analysis of available modelling approaches. Interdependencies, represented by either functional, spatial, or socio-economical linkages, play a dual role in 700 influencing the behaviour of urban systems. On one hand, they serve as a driver of strength, resilience, and control and on the other hand, can also act as a driver of cascading impacts. When disruptions or failures occur in one part of the system, these effects can propagate through interconnected components, triggering a domino effect of consequences that extend beyond the initial point of impact. Such cascading impacts have the potential of leading to system-wide disruptions or failures (Schlumberger et al., 2022).

**Table 4.** Optimization models for recovery planning are grouped into 3 matrices (pre-disaster, pre and post disaster, post-disaster actions) according to the DRM phase where the planned actions should be applied. In each matrix, for each of the selected approaches, it is reported: (i) the urban elements covered by the model; (ii) the goal of the optimization; (iii) if interdependencies are modelled or not (the complete table is available in the supplementary material).

Recovery improvement action planning

| | Author(s) | Urban Element(s) | | | | | | | Objective(s) | | |
|---|---|---|---|---|---|---|---|---|---|---|---|
| | | El | Wt | Gs | Tr | Tc | Gp | Fb | S | T | C |
| **Pre-disaster Actions** | (Y. Fang & Zio, 2019) | ■ | | | | | | | | ■ | |
| | (Der Sarkissian et al., 2020) | | | | ■ | | | | ■ | | |
| | (Du & Peeta, 2014) | | | | ■ | | | | ■ | | |
| | (Rosato et al., 2021) | ■ | ■ | ■ | | ■ | | ■ | ■ | | |
| | (Soltani-Sobh et al., 2016) | | | | ■ | | | | | ■ | ■ |
| | (Bozza, Asprone, & Manfredi, 2017) | | | | | | ■ | | ■ | | |
| | (Y.-P. Fang & Zio, 2019) | ■ | | | | | | | | ■ | |
| **Pre and post disaster Actions** | (Ghorbani-Renani et al., 2020) | ■ | | | | | | | ■ | ■ | |
| | (X. Liu et al., 2021) | | | ■ | | | | | ■ | | |
| | (Ouyang et al., 2012) | ■ | | | | | | | ■ | | |
| | (Ouyang et al., 2019) | ■ | | ■ | | ■ | | | ■ | | |
| | (Blagojević et al., 2022) | | | | ■ | | | | ■ | | |
| | (Almoghathawi & Barker, 2019) | | | | | | | ■ | ■ | | |
| **Post disaster Actions** | (W. Liu et al., 2020) | ■ | | | | | | | ■ | | |
| | (Der Sarkissian et al., 2022) | | | | | | | | ■ | | |
| | (Rosenheim et al., 2021) | | | | | | | | ■ | | |
| | (Cha & He, 2019) | ■ | | ■ | | ■ | | | ■ | | |
| | (Y.-P. Fang et al., 2016) | | | | ■ | | | | ■ | | |
| | (Mudassir, 2020) | | | | | | | ■ | ■ | | |
| | (Almoghathawi et al., 2021) | ■ | | | | | | | | ■ | ■ |
| | (Ghannad et al., 2020) | ■ | | ■ | | | | | ■ | | ■ |
| | (Mudassir & Di Marco, 2021) | | | | ■ | | | | ■ | | |
| | (Bristow, 2019) | | | | | | | | ■ | | |
| | (Guidotti et al., 2019) | ■ | | | ■ | | | | ■ | | |
| | (Cavallaro et al., 2014) | | | | ■ | | | | ■ | | |
| | (Xu et al., 2020) | | | | | ■ | | | ■ | ■ | |
| | (Yu & Baroud, 2020) | ■ | | ■ | | | | | | ■ | |

| Urban Elements | Acronym |
|---|---|
| Electricity network | El |
| Water network | Wt |
| Gas network | Gs |
| Transportation network | Tr |
| Telecommunication network | Tc |
| Good production | Gp |
| Urban facility building | Fb |

| Objective | Acronym |
|---|---|
| Cost minimization | C |
| Recovery time minimization | T |
| Serviceability maximization | S |

| Interdependency | Authors' name background color |
|---|---|
| Interdependency considered | |
| Interdependency not considered | |

Amongst the analysed studies more than 80% of papers published prior to 2019 did not consider interdependencies. Other studies conducted resilience assessments using different indices while considering the recovery phase were explored in our literature study (Aroquipa and Hurtado, 2022; Dong et al., 2021; He and Cha, 2021; Liu et al., 2017; Pant et al., 2014; Zhang et al., 2021), but since they do not address recovery specifically or because they do not clearly discuss the application of their methods in recovery planning, they were not included in Table 4. Also, nine other studies—most of which were conducted before 2019 and which overlooked the interdependencies among systems—are considered in the analysis, which is reported in the supplementary material

Interdependencies can exist among various sectors within an urban system, such as ecosystems, forestry, energy, finance, food, agriculture, etc., leading to the potential emergence of multi-sector risks. Managing intersectoral risks becomes increasingly challenging in geographic contexts prone to multiple hazards, which can amplify cascading impacts across and between different sectors and systems even not directly affected by the hazards (Hochrainer-Stigler et al., 2023). For instance, let's consider a community involved in the recovery process after a disaster. This community relies on importing construction materials from other cities. If a hazard event occurs in the exporting city or along the transportation route connecting the two cities, it can disrupt the supply and demand balance of construction materials. As a result, the recovery process of the community is hindered, impeding their progress toward resilience.

## 5.1 Issue 3 key points

The main key points about the decision-making models and methods for investment in physical elements to improve recovery in urban areas are summarised hereafter:

- Lifeline infrastructure networks are the main physical elements that have been addressed in decision-making models and methods for optimization of investment to improve recovery in urban areas.
- The decision-making models for optimizing investments to improve recovery are mainly concerned with the post-disaster reconstruction time frame.

## 6 Key challenges in multi-risk recovery planning

Stakeholders' decision-making process for selecting an investment direction to increase urban resilience and improve multi-risk recovery planning is complex and often confusing for them. This confusion is exacerbated by the fact that the emerging concepts of resilience, recovery, multi-risk, and built-back-better, among others, have been increasingly included in real-world decision-making processes, but a shared and applicable definition of these terms is still lacking, together with an understanding of the links and relationship among them.

The current challenges and issues in the field of disaster recovery have been extensively analysed in the previous sections of this manuscript, with a specific focus on recovery phenomenology (Sect. 3), its requirements (Sect. 4), and available

methodologies for disaster recovery planning (Sect. 5). In this final section, the key challenges in implementing multi-risk recovery planning are summarized.

Many of the identified challenges are also relevant from a single-hazard perspective, nevertheless, multi-risk conditions can exacerbate or add further complexity to their management. This multi-risk perspective is discussed in detail for seven of the ten identified key challenges. The challenges represent the background to define the required future research development in the field. Each of the ten identified challenges is reported and discussed hereafter.

*1. Current disaster recovery approaches do not evaluate how pre-disaster (i.e., prevention, preparedness, and mitigation)*
*activities can influence the capability of urban systems to recover efficiently and timely from disasters.* Recovery from disasters does not only encompass post-event actions but also pre-event planning, as it is well underlined in some of the definitions of recovery reported in Sect. 3.1. According to the literature, pre-disaster actions play a significant role in determining the speed of the recovery process (Sect. 3.2). Nevertheless, there are only a few approaches in the literature (Sect. 5) that consider both pre- and post-disaster activities as well as how they relate to and affect recovery. These studies are all focused on specific
infrastructures. Since the concept of resilience encompasses a range of time domains from pre-disaster to post-disaster and recovery, an evaluation of the relationship between the different phases of the disaster risk management cycle, considering the activities that are carried on in each phase, is important for taking measures to increase the resilience of the urban system. This issue is not sufficiently addressed so far, specifically by considering the socio-economic aspects of the urban system. Moreover, during the pre-disaster phase, recovery planners often focus on allocating resources and efforts to prioritize single hazards
without considering the potential occurrence of consecutive disasters during the recovery process. The lack of consideration of consecutive disasters can result in increasing the vulnerability and reducing the copying capacity of the population from other types of hazards, by lacking knowledge and equipment to mitigate additional risks that may arise during the recovery period. Furthermore, as it has been depicted in the series of disasters that happened in Japan in 2018 (Sect 4.2), this issue extends to insurance companies as well, as they may not anticipate the possibility of multi-hazard risks and, in the event of
their occurrence, may face unexpected resource constraints in supporting the affected population for recovery and reconstruction.

*2. The recovery process involves a heterogeneous group of stakeholders, characterised by different capabilities, shortages, and expectations.* Decision-making in the recovery process is challenging due to the heterogeneity of the engaged groups specifically in urban areas. Indeed, stakeholders rarely agree on recovery goals (Sect. 3.3). If the different options in defining
the objectives for the recovery procedures are not considered and the participation of all stakeholders is not promoted, the outcomes could be detrimental to one or more groups, leading to greater inequality in the community. Stakeholders' heterogeneity can be seen as a resource in the different stages of the recovery planning (e.g., needs and capacity assessment, or the implementation of the recovery actions) since it would bring different skills, mindsets, and capabilities which would enrich and improve the overall process (Sect. 4.1).

*3. Rather than viewing recovery as a process, it is mainly addressed by focusing on the outcome.* Most definitions of recovery, even the less recent ones, referred to disaster recovery as a process (Sect. 3.1). However, it emerged from experience that while

planning for recovery, the main emphasis was on the outcomes. This issue has led stakeholders to think about recovery mainly out of a static mindset, even in identifying the needs and scopes (Sect. 4.1). Moreover, stakeholders usually compare the states of the system before the disaster and at the end of the recovery process, while it has been outlined that it would be better to

compare the trends that the system exhibits in each time domain to determine whether the system was improved or not (Sect. 3.3). Therefore, even though the need to view recovery as a process rather than a product has been highlighted in theory, this approach mostly has not been executed in practice. Moreover, avoiding considering recovery as a process developing in time, might lead to disregarding the changing vulnerabilities of the community after a disaster and throughout the recovery process. Consequently, this neglect may lead to insufficient preparation and planning for potential hazards that may happen during the

recovery (Sect. 4.2).

*4. Physical reconstruction has been the focus of recovery plans.* The planning for other recovery activities that support the restoration of urban socio-economic institutions has been only partially explored in the literature. Reconstruction and recovery are different even in definition (Sect. 3.1). Insufficient research has been conducted on the interaction between physical reconstruction and socioeconomic recovery efforts, as well as the coordination of these two types of activities so that they

would be carried out with the same objectives and in line with each other (Sect. 3.1).

*5. The current literature does not establish a clear relationship between the response and recovery phases.* Communities in these two phases have different needs and goals. Additionally, from response to recovery, capacities and external contributions may change significantly. Therefore, the transition between the two phases needs to be managed effectively so that stakeholders and decision-makers would not become confused by abrupt changes (Sect. 3.2). Moreover, it could happen for a community

to be in the recovery phase from a disaster while in the response phase for another. From a multi-risk perspective, the DRM cycle framework has shown limitations in addressing simultaneously response and recovery activities. In response to these shortcomings, researchers have proposed new frameworks that aim to capture the complexities involved. However, it is worth noting that limited effort has been put to transfer these frameworks into executive guidelines and plans. (Sect 3.2) to ensure that planning and actions in response and recovery phases should be coordinated to not conflict with one another (Sect. 4.2).

*6. Recovery is a dynamic and non-linear process, characterised by different paces and parallel activities.* Different communities and even different groups within the same community may experience varying rates of recovery, depending on their equipment capabilities, willingness to participate in the recovery process, and recovery objectives (Sect. 3.3 and 4.1). In addition, past experiences indicate that recovery efforts would not be carried out in every community in the same pattern and order, since the built environment components could be damaged at different levels depending on their significance to the

functioning of the urban system. Therefore, it is not a realistic approach to prescribe and demonstrate an ordered linear recovery roadmap for all communities (Sect. 3.2 and 2.1).

*7. Disasters are not seen as an opportunity to improve the urban system's resilience.* Planning and investments for recovery are concentrated mainly on getting the system back to its pre-disaster state (Sect. 3.3 and 3). There are not enough plans and strategies that view recovery from disasters as a springboard for achieving sustainable development goals or making cities

more resilient. The BBB concept has been in existence for years, but it is still ambiguous. All the components of this concept

('build', 'back', and 'better') are still controversial and there is disagreement amongst stakeholders on a practical definition of the terms (Sect. 3.3). Moreover, the existing literature on the concept of BBB lacks sufficient integration of the multi-(hazard-)risk perspective. In practical terms, during the recovery process, the emphasis is predominantly placed on addressing the hazards that have recently affected the urban area and improving its resilience in relation to those specific hazards. However, other potential hazards that have not recently impacted the recovering area, even though they may pose significant risks, are not considered when implementing DRR measures during the recovery process (Sect 3.3).

*8. Community needs are not always incorporated into the recovery planning and reconstruction process.* Therefore, the physical reconstruction of a built environment after a disaster is not always compatible with stakeholders' expectations and needs and eventually results in the malfunctioning of the system. Needs are changing constantly and answering one need leads to the emergence of another (Sect. 4.1). Furthermore, effectively managing the balance between addressing community needs and enhancing resilience against hazards beyond those that have caused damage from a multi-risk perspective, requires the incorporation of co-production knowledge processes involving stakeholders and the scientific community. By involving these key stakeholders, all aspects of multi-risk resilience can be addressed, including those that may not be readily apparent to the local community and decision-makers (Sect 4.1).

*9. Multi-risk recovery planning approaches are still lacking.* Neglecting synergies, cumulative damages, recovery dynamics, and any other interactions at different levels (hazard, vulnerability, exposure, or DRR measure) might result in recovery plans that would strengthen the system's resilience regarding a particular hazard but leave it vulnerable, or even increasing its vulnerability to other hazards (Sect. 4.2).

*10. Recovery planning requires a multi-objective optimization including maximisation of the socio-economic benefit of the community.* The socio-economic importance of the urban assets and their contribution to the overall system functionality has been neglected in the planning and orientation of the investment for the recovery process. Only cost and time minimization and in some cases maximising serviceability of specific infrastructures were the main objectives in current optimization models. Moreover, socio-economic interdependencies amongst urban assets (e.g., buildings, critical infrastructures, lifelines, etc.) are ignored so far in urban system modelling for recovery (Sect. 5)**.** In an interdependent system hazard negative effects can propagate beyond their initial point of impact. This can lead to multi-risk systemic impacts that could occur even though two or more hazards do not overlap in space and time. These types of interdependencies have not been addressed in available optimization models.

## 7     Conclusions

In this paper, a critical review of the existing natural hazard disaster recovery literature and guidelines has been performed with the dual goals of identifying current challenges and laying the groundwork for future research on multi-risk recovery planning for urban resilience and decision-making tools development.

Disaster recovery literature has been investigated with a specific focus on: how we can define recovery and its role in the risk management cycle and what is the final goal of the recovery process (Issue 1, Sect. 3), what are the most important prerequisites for the urban system to begin and sustain recovery in a multi-(hazard-)risk environment (Issue 2, Sect. 4), and what are the available disaster recovery planning models and methods (Issue 3, Sect. 5). What emerged from the review is that disaster recovery has been largely addressed by different sectoral perspectives and scientific communities. Nevertheless, studies providing holistic approaches to recovery, not only focusing on physical reconstruction but also including socio-economic aspects, are still lacking. Furthermore, recovery is mainly approached from a single-risk perspective, neglecting synergies, cumulative damages, recovery dynamics, and any other multi-(hazard-)risk interaction. Moreover, recovery has been only marginally explored from a pre-disaster perspective, in terms of planning and actions for better recovery before disasters occur. Multi-risk recovery planning requires a multi-objective optimization, where not only time and money are minimised, but also the socio-economic benefit of the community is maximised. Nevertheless, community needs are not always incorporated into the recovery planning and reconstruction process, and stakeholder heterogeneity is not exploited as a source of richness, but as a limit in identifying optimal solutions.

Stakeholders' decision-making process for selecting an investment direction to increase urban resilience and improve recovery planning is still complex and confusing. This confusion is exacerbated by the fact that applicable definitions of 'resilience' and 'recovery' are missing, together with an understanding of the links and relationship among them, and a clear distinction between response and recovery phases is still lacking, while communities in these two phases have different needs and goals. Furthermore, planning and investments for recovery are concentrated mainly on getting the system back to its pre-disaster state. There are not enough plans and strategies that view recovery from disasters as an opportunity to improve the urban system's resilience and BBB. Moreover, the main emphasis in recovery planning is still on the outcome, without considering recovery as a dynamic and non-sequential process, characterised by different paces and parallel activities. This misperception led the scientific community to suggest an ordered sequential recovery roadmap for all communities, that is not able to successfully reflect the real-world dynamics.

The outcomes of this critical literature can set the basis to outline the key research directions in the field of disaster recovery and urban resilience. Nevertheless, it is important to note that while all the challenges outlined primarily pertain to urban areas, some of them could also be applicable to the recovery processes in other contexts, such as rural settlements. As a key direction, the pre-disaster time domain should be considered as the beginning point of the recovery process, using the time before the occurrence of a disaster to plan, but also to take actions to get the built environment prepared for a good, advantageous, and quick recovery. Furthermore, future studies should model the recovery process in a dynamic way rather than with a series of sequential actions. In such a way decision-makers and planners will be able to implement early recovery activities and developmental reconstructions simultaneously to maximise the functionality of urban systems in the shortest time possible. Recovery planning should be addressed by developing holistic tools that consider the relationship between the different DRM phases and the impact of the activities in each phase one on another, to better optimise the investments with the goal of increasing urban system resilience. Future recovery planning approaches should promote a functional recovery, where physical

reconstruction and socio-economic recovery are jointly achieved. To ensure such an ambitious result, one of the focal points of future recovery research should develop strategies for enhancing and facilitating participatory recovery planning and actions, to ensure that stakeholders' needs, requirements, viewpoints, and preferences are successfully included. The future recovery planning models and tools would be general and flexible enough to fit different urban systems with varying socio-economic

characteristics, stakeholder preferences, and exposure to multiple hazards. Moreover, disaster recovery research should permanently shift from a single to a multi-risk perspective, providing comprehensive tools for recovery planning able to capture asynergies, cumulative damages, and multi-risk recovery dynamics. The identified research direction will ultimately enable stakeholders in making decisions and optimising their investments in the pre-disaster phase to improve the urban system's recoverability and overall urban resilience.

**8    Author contribution**

S. M.: Conceptualization, Methodology, Formal analysis, Visualization, Writing - Original Draft, Writing - Review & Editing.

S.D.A.: Conceptualization, Methodology, Writing - Original Draft, Writing - Review & Editing.

S.C.: Conceptualization, Supervision, Writing - Review & Editing.

G.B.: Conceptualization, Supervision, Writing - Review & Editing.

F.P.: Conceptualization, Supervision, Writing - Review & Editing.

All authors have read and agreed to the published version of the manuscript.

**9    Competing interests**

The authors declare that they have no conflict of interest.

**10    Acknowledgements**

The study presented in the paper was partially developed within the research activities carried out in the frame of 2022-2024 ReLUIS Project – WP4 Seismic Risk Maps (MARS2) (Coordinators: Sergio Lagomarsino and Angelo Masi) and partially within the RETURN Extended Partnership. The ReLUIS Project is supported by the funding of the Italian Department of Civil Protection while the RETURN Extended Partnership received funding from the European Union Next-GenerationEU (National Recovery and Resilience Plan – NRRP, Mission 4, Component 2, Investment 1.3 – D.D. 1243 2/8/2022, PE0000005). Note

that the opinions and conclusions presented by the authors do not necessarily reflect those of the funding entities.

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
