# Peer review of "Review article: Current approaches and critical issues in multi-risk recovery planning of urban areas exposed to natural hazards"

_EGUsphere, 2023_

## Author Comment (AC2)

**Review article: current approaches and critical issues in multi-risk recovery planning of urban areas exposed to natural hazards - Soheil Mohammadi et al., 2023**

**Reply to reviewers' comments**

**Reviewer #1 (Marleen de Ruiter)**

**General comments**

**(C1)** I think the paper could benefit from some restructuring.

> **(C1.1)** There are many 1-3 sentence paragraphs, which seem to interrupt the flow of the paper and at times cause some confusion. Examples include, but are not limited to:
>
> 90 onwards: this seems to still be about the methods (critical lit review) so should this paragraph be part of the previous paragraph (L. 86-83), which is also on the critical lit review?
>
> 147-151: should this paragraph maybe be part of the previous and/or next paragraph?
>
> **[Authors reply]** To improve the flow of the text, we checked the overall manuscript and reorganized the sentences into coherent paragraphs, avoiding having too small (e.g., 1-3 sentence) paragraphs. These changes include but are not limited to the two examples you suggested.
>
> **(C1.2)** To me it seems that paragraphs L.46 – 54, L. 55-62, and L. 67-78 discuss definitions of / changes in thinking about resilience and recovery while in between those, L.63-66 seem to introduce the urgency to adopt a multi-risk perspective.
>
> **[Authors reply]** To improve the logical coherence of the text and keep together all the paragraphs investigating the definitions of / changes in thinking about resilience and recovery, we moved the concepts reported in L. 67-78 just after the ones in L.46 – 54 and L. 55-62. Moreover, we made some changes in the paragraphs (L.46-78) to improve their readability and the logical flow among them. The text flow appears as it follows:
>
> *"Urban resilience has been defined variously in the literature. Among the many different definitions of urban resilience available in the literature, one of the most*

*comprehensive is the one provided by Meerow et al. (2016),: "Urban resilience refers to the ability of an urban system and all its constituent socio-ecological and socio-technical networks across temporal and spatial scales to maintain or rapidly return to desired functions in the face of a disturbance, to adapt to change, and to quickly transform systems that limit current or future adaptive capacity." The phrase "ability to return"stands out in this definition. The fundamental concern that arises, however, is how to ensure that the urban area , as a complex system with various components and interconnected networks, has the ability to return or recover after a disaster occurs. On the other hand, the term "maintain" can be seen in Meerow et al. (2016) definition as a reference to the ability of a resilient urban system to preserve what currently exists, which is more closely tied to the concepts of disaster risk preparedness and response.*

*As pointed out by Rus et al. (2018), the resilience concept can be reflected in preparedness, response, recovery, and adaptation actions, depending on the temporal domain. However, research that evaluates the relationships and implications of these different actions on one another in resilience-based urban planning are still absent (Rus et al., 2018). This lack in the literature is reflected also in the confusion felt by stakeholders involved in the resilient urban planning process. Indeed, decision-makers do not know how to choose an investment direction among the various phases of the Disaster Risk Management (DRM) cycle (Kawasaki and Rhyner, 2018). Are they supposed to protect and strengthen the system, work to make it more recoverable, or even set aside funds for disaster recovery?*

*Although the resilience of an urban system is not solely determined by its ability to restore from a disruptive event, the recovery process represents one of the most critical and significant aspects contributing to the overall system resilience. To corroborate this perspective, Manyena et al. (2019) indicate that, since 1980 the terms "return to equilibrium," "bounce-back," "recover," "restore," "bounce-forward," "rebound," "rebuild," and "reorganize", which all have the connotation of recovery, have been frequently used in the different resilience definitions provided in the literature. According to McEntire et al. (2002), the term "resilience" emerged as a reaction to or alternative for the term "resistance". The key difference between these two concepts is that contrary to the resistance idea that prevention is the main strategy, natural hazard events inevitably occur in resilience discourse and disaster avoidance is not always achievable. As a result, the main issue in resilience is the need to focus on recovering from disasters as quickly and effectively as feasible (McEntire et al., 2002). Indeed, different researchers have used indicators such as the system's ability to function during recovery, the speed of recovery, the quality of recovery, and the area under the recovery curve to measure the system's resilience (Bruneau et al., 2003; Rus et al., 2018; Soltani-Sobh et al., 2016; Zhang and Wang, 2016). Although recovery is a fundamental aspect that is well captured by the resilience concept definition, this DRM phase is the least explored in the literature in comparison to the others (Der Sarkissian et al., 2021; Rodríguez et al., 2018)."*

The concept provided on L.63-66 about the urgency to adopt a multi-risk perspective, has been moved just after this part and enriched significantly (***see answer to comment C.2.***)

**(C1.3)** The authors could consider moving the method description (second half of the intro) to a separate method section (basically, including a section header at the start of the description of the approach).

**[Authors reply]** Thank you for this valuable suggestion. We created a separate section (Sect. 2, Methodology) where we moved the text originally reported in L.86-131. This change has allowed us to provide more visibility to the methodology on one side, but also to lighten the introduction. This last aspect has allowed us to strengthen the multi-risk aspects of the introduction by adding some dedicated paragraphs(see also reply to your next comment). Moreover, as suggested by the other reviewer, we also included in the new section 'Methodology' a figure (Fig.1), where we graphically depicted the relationship between the identified Issues and the guiding Research Questions. The figure is attached to this document as an annex (**Annex 1**).

**(C.2)** I think the multi-risk aspect of the paper is an important novelty and strength of the paper. I would encourage the authors to strengthen this in their introduction. In the paragraph at L. 63-65 the multi-risk aspect seems to almost appear as an afterthought. I encourage the authors to strengthen this particular element of the paper in the introduction, including a more elaborate explanation of those "additional challenges in decision-making". In restricting the intro (see previous point), this particular element could maybe be described more prominently and extensively. I can imagine something more general on multi-risk recovery challenges and those specifically within an urban setting.

**[Authors reply]** We provided more context about the multi-risk challenges in disaster recovery and the importance of adopting a multi-risk perspective in this realm, by adding some additional paragraphs in the introduction.

*"When urban systems are affected by complex disaster scenarios, involving potential impact interactions, addressing the recovery process can become more complicated. In that circumstance, the system must be resilient to many types of risks, embracing a multi-risk perspective, which will introduce additional challenges in the decision-making process regarding the recovery (Curt, 2021). Multi-(hazard)-risk, as collectively named by Ward et al. (2022), encompasses all disaster risk assessment and management approaches that consider interactions or interdependencies among different hazards, vulnerabilities, or risks. These approaches are able to better capture complex risk dynamics which are*

*increasingly impacting urban areas worldwide. The interrelationship of multiple hazards and their impacts, as well as the implications of DRM decisions on different economic sectors and regions, and the diverse impact of disaster risk reduction measures on different risks, make recovery in multi-hazard environments challenging (Ward et al., 2022; Hochrainer-Stigler et al., 2023). Among the different types of multi-hazard interaction mechanisms that can lead to a disaster such as compound, triggering, or cascading, (Gill and Malamud, 2014; Marzocchi et al., 2009; Tilloy et al., 2019), the occurrence of consecutive disasters is specifically challenging for the recovery process. Consecutive disasters are two or more disasters that occur in succession and whose direct impacts overlap spatially before the recovery from the prior event is considered complete (de Ruiter et al., 2020). The results of the interaction between the impacts generated by two consecutive hazards depend on the time interval between them, the rate of recovery of the system or the asset, or a combination of them (De Angeli et al., 2022; Marzocchi et al., 2012)."*

Moreover, we supported these concepts by providing a real-world example that highlighted the complex challenges posed by consecutive disasters in recovery:

*"As a real example in the western part of Iran, a devastating earthquake occurred on November 12, 2017, at the Iran-Iraq border, causing the death of at least 630 people (Naserieh et al., 2022). The recovery efforts began by providing temporary shelters to the more than ten thousand affected population (Omarzadeh et al., 2021). Prior to the earthquake, the country was facing a prolonged period of drought, and no one anticipated the possibility of a flood occurring within some months (Yadollahie, 2019). However, from mid-March to April 2019, widespread flash flooding occurred, affecting large areas of Iran, including the regions that were undergoing recovery from the earthquake (Miri et al., 2023). The potential occurrence of a flood was not taken into account, leading to the establishment of temporary shelters alongside canals, which resulted in the flooding of the people residing in those shelters and imposed significant economic impacts on them."*

The multi-risk concept has been also strengthened in other sessions on the manuscript, by adding some dedicated paragraphs inside the discussion of Issues 1, 2, and 3 ***(see answer to comment C.11.b).***

**(C.3)** 81: "inadequacies" in terms of how we define post-disaster recovery or how we manage post-disaster recovery?

**[Authors reply]** The sentence reported in line 81 was unclear and the word "inadequacies" was not informative enough. We rephrased it as follows:

*"This study analyses the existing disaster recovery literature and guidelines in the realm of natural hazards with the dual goals of (i) identifying current issues in multi-risk recovery planning for urban areas and (ii) laying the groundwork for developing multi-hazard decision-making planning tools for recovery."*

**(C.4)** 113-114: how were these particular key words selected?

**[Authors reply]** The selected keywords (recovery requirements, post-disaster needs, resilient recovery, the difference between recovery and emergency, multi-risk recovery, recovery optimization models, recovery planning, pre-disaster recovery planning, socio-economic aspects of disaster recovery) have been selected to be linked and representative of the four identified Research Questions. We specified this aspect in the Methodology (Sect.2), by adding the following sentence:

*"To address the specific Research Questions, the following keywords were selected: difference between recovery and emergency, pre-disaster recovery planning (Question 1); resilient recovery, socio-economic aspects of the recovery (Question 2); recovery requirements, post-disaster needs, multi-risk recovery (Question 3); recovery optimization model, recovery planning (Question 4)."*

**(C.5)** Reading L. 132-133, it is unclear to me how this is different from what will be addressed in Issue 2.

**[Authors reply]** Section 5 is reporting the key findings from the critical literature review that is discussed in Sections 2 to 4, while Issue 2 (Sect. 3) is addressing a specific aspect, i.e. the recovery requirements in the current world setting.

Nevertheless, we realized that the title we provided for Sect.3 and also the description of Issue 2 in L. 126-127 is misleading and is creating confusion with the title provided for Sect. 5.

To avoid misunderstandings, we removed the word 'challenges' from the title of Sect.3 as follows:

*Recovery requirements in the current world setting*

Moreover, we also modified the text at L.126-127 in the following way:

*Issue 2: Recovery requirements in the current world setting*

**(C.6)** Table 1 is a great way to demonstrate the evolution of our understanding of "recovery" over time. Very nice!

**[Authors reply]** We are really pleased you appreciated the table. The aim was exactly, as you wisely noticed, to better illustrate how the recovery concept has changed over time, moving to a more holistic definition.

**(C.7)** 169: it may be helpful to include a definition of build back better. This may for example help to understand L. 176. In addition, I found the paragraph of which it is part (L.165 – 179) a bit difficult to follow (for example, what do the authors mean by the "subjective functions" and "objects").

**[Authors reply]** We agree that a definition of Build Back Better (BBB) should be provided to better understand L.176. For this reason, just after the first time the BBB has been mentioned in the manuscript, we added the following:

*According to the UNDRR (2020), BBB is defined as "the use of the recovery, rehabilitation and reconstruction phases after a disaster to increase the resilience of nations and communities through integrating disaster risk reduction measures into the restoration of physical infrastructure and societal systems, and into the revitalization of livelihoods, economies and the environment"."*

Moreover, we tried to simplify L.165 – 179, avoiding the use of the terms "subjective functions" and "objects". Please, find the rephrased paragraph hereafter:

*"The fundamental distinction between the two definitions of 'recovery' and 'reconstruction' provided by UNDRR lies under the fact that the 'reconstruction' definition refers to more tangible elements. On the other hand, in the 'recovery' definition, a broader array of functions, (sub)systems, and activities of a community or society, which also encompass intangible elements, are mentioned. This emphasises the fact that recovery encompasses more than the rebuilding of the physical components of the systems. In addition, while both definitions of reconstruction and recovery from UNDRR contain advice about aligning with BBB principles, the reconstruction definition refers to the goal of reaching a full community functioning, while the recovery definition emphasises more on the opportunity for 'improvement'."*

**(C.8)** 261: I think it is very nice that the authors mention an example that shows the fuzziness of the boundaries between stages. Maybe they can describe this in a bit more detail to demonstrate the fuzziness more clearly to people who are less familiar with the aftermath of this particular disaster?

**[Authors reply]** We provided more details about what happened in the aftermath of this devastating earthquake and we provided additional information to better highlight the fuzziness between early recovery and the development phase of the recovery:

*"The recovery in L'Aquila following the earthquake in 2009 is a good illustration of how the boundaries between the various stages of recovery are not always clear and can be fuzzy. In April 2009, L'Aquila, Italy was hit by a 6.3 MW earthquake with a hypocentral depth of 10 km. Reconstruction efforts led to the construction of 5722 housing units in 19 new settlements distributed around the city within one year of the earthquake (Contreras et al., 2013). The newly built housing included features like seismic isolation and solar cells, typically associated with the development phase of recovery. However, by 2014, damaged houses could still be found in the city center, indicating the presence of the early recovery phase. The simultaneous presence of early recovery and development recovery actions highlights the fuzziness boundaries between different phases of post-disaster recovery (Contreras, 2016; Contreras et al., 2014)."*

**(C.9)** At the start of section 2.2, it may be nice to remind the reader of the phases of the DRM cycle.

**[Authors reply]**  We added a couple of sentences at the beginning of section 2.2 to introduce the main phases of the DRM cycle:

*"The DRM cycle is widely recognized in the global DRM community as a framework for managing various types of disasters, both natural and anthropogenic (Coetzee and Van, 2012). It is characterized by separate and sequential phases with varying durations and actions. While phases' number and their naming vary in the literature, the following main ones (before, during, and after the event) can be identified: preparedness and mitigation, response, and recovery."*

Moreover, according to the suggestions of the other reviewer, we highlighted the difficulty of the current DRM cycle to capture the dynamics and interaction of multiple hazards, particularly those involving both sudden-onset and slow-onset hazards, and we mentioned alternative DRM models, such as the ones recently proposed by Bosher et al. (2021), Staupe-Delgado (2019) and Terzi et al. (2022).

**(C.10)** Since some sections are lengthy and contain a lot of information, the authors could consider adding a couple of sentences at the end of each issue bringing it together.

**[Authors reply]**  To facilitate the reader in keeping some "take-home messages", we added a dedicated section at the end of each issue, where we summarized a few key points each.

**Issue 1 key points**

- The concept and definition of recovery have evolved over time passing from merely focusing on physical reconstruction towards inclusion of socio-economic aspects of recovery in the definition.
- The distinction between emergency and recovery phases is still a controversial issue and has been the motivation for defining the transition phase between them.
- Recovery planners typically view recovery as a sequential and predefined process, divided into subphases. However, real experiences challenge this perspective, as the recovery process is often more complex and unpredictable.
- Determining the ultimate goal of recovery, such as achieving resilience through a BBB approach to prevent future disasters, adds complexity to recovery planning, especially in multi-risk environments.

**Issue 2 key points**

- The post-disaster needs and priorities of the population, which should be at the core of recovery planning, vary across different social and geographic contexts.
- The needs of the population change over time in the aftermath of a disaster.
- Community involvement might occur both in the needs and capacity assessment phase to tackle the problem at the community level and efficiently employ local resources.
- Current approaches to multi-risk recovery planning only partially address the complexity of such topic, since: (i) they focus on hazard prioritization ignoring vulnerability dynamics exacerbated by consecutive disasters; (ii) they do not take into account potential asynergies between DRR measures, as well all damage accumulations; (iii) they do not evaluate systematically the interdependencies amongst different components of the urban system.

**Issue 3 key points**

- Lifeline infrastructure networks are the main physical elements that have been addressed in decision-making models and methods for optimization of investment to improve recovery in urban areas
- The decision-making models for optimizing investments to improve recovery are mainly concerned with the post-disaster reconstruction time frame

**(C.11.a)** Since section 5 aims to focus on challenges specific to multi-risk planning (in urban areas?), I wonder if it would be possible to focus that section and the identified challenges specific to multi-risk (in urban areas?). As I understand it now, some of these challenges are not unique to multi-risk recovery but rather recovery in general.

**[Authors reply]** We agree that the challenges reported in Sect.5, as a result of the summary of the main outcomes discussed in Sects. 2-4, were not focusing enough on the multi-risk aspect. Therefore, we added additional sentences referring more specifically to multi-hazard aspects in 6 over 9 of the remaining challenges (one of them was already multi-risk oriented). Please, find in **Annex 2** the new version of the challenges, with the additional text reported in red.

Regarding your observation about the fact that some of these challenges are not unique to multi-risk recovery but rather recovery in general, this is correct indeed. Many of the identified challenges are relevant also in a single-risk perspective. Nevertheless, multi-risk can exacerbate them. We highlighted this concept at the beginning of Sect. 5, adding the following paragraph:

*"Many of the identified challenges are also relevant from a single-hazard perspective, nevertheless, mult-risk conditions can exacerbate or add further complexity to their*

*management. This multi-risk perspective is discussed in detail for seven of the ten identified key challenges. "*

**(C.11.b)** This is maybe something the authors could also reflect on in the earlier sections; whether these sections can be tailored more to multi-risk?

**[Authors reply]** According to your main suggestion, we strengthen the focus of the overall manuscript on the multi-risk aspect. We added some dedicated paragraphs in the introduction ( *see answer to comment C.2.*)

Moreover, we added some dedicated paragraphs inside the discussion of Issues 1, 2, and 3 discussing more in detail:

- the difficulty of keeping DRM phases separate when dealing with multi-risk and emerging DRM models alternative to the traditional cyclic approach (Sect. 2.2)
- the exploitation of the build-back-better concept from a multi-risk perspective (Sect. 2.3)
- how multi-hazard conditions can exacerbate the communities' dilemma in balancing the need to enhance resilience against future disasters and the desire to quickly rebuild their houses and livelihoods (Sect. 3.1)
- multi-risk multi-sectoral challenges related to interdependencies among urban system components (Sect. 4)

As a consequence of this spread of multi-risk related criticalities along the manuscript, we also reorganized more effectively the section originally devoted to multi-risk (Sect. 3.2). More specifically we focused it more on the recovery requirements seen from a multi-risk perspective and we consequently renamed it as "Multi-hazard risk recovery requirements". Furthermore, we moved some of the paragraphs into the Introduction, and added some paragraphs covering the following aspects:

- multi-risk prioritization in recovery planning
- asynergies of DRR measures implemented during reconstruction and recovery
- impact of multi-risk scenarios on the allocation of resources for recovery activities

**Minor comments**

**(C.12)** Some writing issues. E.g. the first sentence of the abstract: there is a strong push within the field to move away from using "natural disasters". Here I think it could be easily avoided by writing "Post-disaster recovery…".

**[Authors reply]** We replaced "natural disasters" with "post-disaster" in the abstract.

**(C.13)** In the third sentence of the abstract: do the authors mean planning (that takes place before a disaster) of recovery-related actions (e.g., getting a contract for debris cleaning) or do they mean preparedness actions (e.g. to mitigate the potential impacts of a disaster).

**[Authors reply]** The intended meaning of the sentence encompasses more than just recovery planning or soft action; it also emphasizes the need for proactive measures to enhance the recoverability and resilience of urban areas. Therefore, in addition to preparedness actions, the sentence highlights the significance of implementing changes in pre-disaster periods, particularly through modifications in physical structures and infrastructures, to augment the recoverability of urban areas. The underlying concept of the sentence revolves around the notion that recovery efforts should not be restricted to immediate response and recovery but should encompass long-term measures aimed at strengthening the capacity of urban areas to recover and bounce back or bounce forward from adverse events. For more clarification, we changed the sentence to:

*"Furthermore, recovery has been only marginally explored from a pre-disaster perspective in terms of planning and actions to increase urban resilience and improve urban systems recoverability."*

**(C.14)** 26: I assume the time stamp here is a typo?

**[Authors reply]** The time stamp was a typo and has been removed.

**(C.15)** 28 "dates" back to?

**[Authors reply]** It is modified to:

*"back to 1973"*

**(C.16)** 31: it is a bit unclear what is meant here with "complex system". In general, the authors use many different disaster related terms (e.g., natural hazard event, extreme event, disturbance, etc); this can create some confusion. The authors could consider adding a box or table defining some of the key terminology used in their paper.

**[Authors reply]** We included a definition of "complex system" and "disturbance" along the text as reported hereafter:

*"Holling's writing was the basis for developing the resilience concept and applying it to understand the performances of complex systems, i.e. systems "in which there are multiple interactions between many different components" (Rind, 1999), when encountering disturbances (Walker et al., 2004; Gunderson and Holling, 2002). According to an ecological science perspective, disturbances can be seen as massively destructive and rare events (Rykiel, 1985) that impact the system from the outside. "*

Regarding the term 'extreme event', this was mentioned only once (L.24). To avoid misunderstandings we removed it from the sentence, which is now written as: *"To cope with natural hazard event impacts, planning and procedures that anticipate their occurrence, mitigate possible damages, and allow for the speedy restoration of key services and recovery are required (Berke et al., 2009)."*

**(C.17)** 37-38: I am not sure I understand the second half of the sentence; it is not clear to me how the previous statement (or definition) supports the claim in this sentence that resilience is linked to DRM.

**[Authors reply]** Your observation is truly correct. We modified the sentence eliminating the link to the DRM:

*"As one of the most complex systems exposed to natural hazards, urban areas have also been included in the resilience concept's extent."*

**(C.18)** 47: I believe that with a quote, the citation also need to include a page number – but I may be wrong there.

**[Authors reply]** Thank you for noticing that. We carefully checked it and we found that it is not required according to the "English guidelines and house standards" section in the NHESS Submission instructions.

**(C.19)** 63: "a variety of sorts and categories" -> not sure what the authors mean here. Could they be more explicit?

**[Authors reply]** The sentence was removed entirely by answering other comments (specifically the one related to strengthening the multi-risk perspective in the manuscript) and reordering the paper.

**(C.20)** 85: maybe include a "(e.g., XXXX)" to explain what you mean by physical elements (buildings, infrastructure,...?)

**[Authors reply]** The sentence was modified to:

*"In this research, we focus on decision-making concerning investment prioritization to improve the resilience of physical elements (e.g., structures, buildings, infrastructures, open spaces, etc.) at the urban scale."*

**(C.21)** 130-131: maybe for legibility, the authors could consider adding this info to the issue 1 -3 description in L.125-129 instead of having a separate sentence on this.

**[Authors reply]** Your observation is truly correct. For increasing legibility, we changed the format of the numbers of the questions and the correspondences amongst questions and issues:

*Issue 1: Recovery and its role in the risk management cycle (Sect. 3) that corresponds to Research Questions 1 and 2.*

*Issue 2: Single and multi-risk recovery and requirements in the current world setting (Sect. 4) that corresponds to Research Question 3.*

*Issue 3: Decision making models and methods for investment in physical elements to improve recovery in urban areas (Sect. 5) that corresponds to Research Question 4.*

**(C.22)** 140-142: "recognizable activities" and "evolutionary notions" not sure what the authors mean.

**[Authors reply]** The expression *"recognizable activities"* has been changed to *"pre-defined activities"*, to highlight how recovery was seen as a static and standardized process.

The term *"evolutionary notions"* has been replaced with *"progressive concepts"*, since it was referring to the inclusion of some new ideas (i.e. risk reduction, decreased vulnerability) in the original definition of recovery provided by Haas et al. in 1977.

**(C.23)** 208-211: I wonder if there is any literature to support these statements.

**[Authors reply]** Five references were added. Now the text it appears as follows:

*"As shown in panel (b) of Fig. 1, by increasing the intensity of socio-economic recovery activities, the common zone (functional recovery) could be preserved or even increased during the recovery period. Furthermore, the intensity of socio-economic recovery activities could be increased without external support in a disaster-struck community (Alifa and Nugroho, 2019). These cooperatively evolving activities enable people to take part in the restoration of their communities independently (Nigg, 1995; Talbot et al., 2020; Perce, 2007). For instance, as more enterprises of all sizes become involved in the economy, people will be more capable of actively participating in the economic recovery of their community (Freeman, 2004). Setting this balance between physical reconstruction and socioeconomic recovery would be possible if the disaster area needs assessment (see Sect. 3) is considered."*

**(C.24)** 219: what is meant by "diverse groups"?

**[Authors reply]** It has been modified to:

*"social classes, races, ages, genders, and family statuses" according to Cutter et al. (2006) and Fussell (2015).*

**(C.25)** Figure 2: looks great, very useful. Maybe the authors can add BBB to the legend or description and explain its meaning (in relation to Point F). Maybe the font size of the x and y-axis labels could be decreased a little.

**[Authors reply]** We have modified the figure accordingly. Here is the updated version:

[Figure]

**(C.26)** 485-489: the use of both multi-risk and multi-hazard may cause some confusion.

**[Authors reply]** In order to prevent potential confusion and to provide a more comprehensive representation, we have made the decision to revise all the terms to "multi-hazard risk" throughout this paragraph.

**(C.27)** Figure 4 is again; I was just a bit confused by the use of different DRM phases compared to those presented in figure 2. It may be helpful to the reader to add the meaning of ABC to the figure caption.

**[Authors reply]** We appreciate your feedback and valuable input. In Figure 4, our intention was to depict the functionality of an urban system across different temporal domains in relation to the occurrence of a disaster. In order to avoid any potential confusion, we have made the necessary amendment of changing the term "phases" to "time frames" in both the figure and its description. We also changed the colors of the time frames to create a link between them and the DRM phases reported in figure 2. Here is the updated version of the figure:

[Figure]

We intentionally used the "time frames" in Figure 4 instead of the DRM phases (as we did in Figure 2) to present a broader perspective. We intentionally chose not to strictly adhere to the traditional DRM phases in order to encompass a wider range of actions taken during the pre-disaster time frame. By doing so, we sought to include not only preparedness and mitigation actions, but also other forms of actions that have an impact on the recovery process during the pre-disaster phase.

For demonstrating the meaning to A, B, and C panels we added the below paragraph to the caption of the figure:

*"Distinct variations in recovery rates are discernible across the white panels. Panel A illustrates the recovery process following a single rapid onset hazard. In Panel B, recovery ensues after a slow onset hazard, disregarding other potential hazards, resulting in a system vulnerable to other hazards such as floods. Nevertheless, the subsequent recovery rate surpasses the rate of the recovery after the initial rapid onset hazard, namely the earthquake. Panel C highlights a segmented decline in system functionality, attributed to inadequate recoverability in the face of consecutive events happening within short intervals."*

**(C.28)** In general, and as pointed out by the other reviewer, the paper could benefit from an English language check (sentence structure, word use, missing words, etc). I highlight some things but these are just some examples:
- 28 "dates" back to?
- 57: "DRM cycle phases one on another" -> "on one another"
- 75: "recovering from them" -> from disasters?

- Incorrect use of "the" (e.g., L. 93: 'the recovery" -> recovery; L. 111 "the google scholar" -> google scholar)
- 194: "However, this would not be possible." -> this is never possible or this is challenging?
- 197: "because socioeconomic recovery is dependent on physical structure" -> is dependent on the recovery of physical structures (?)
- 208: "in a disaster community" -> in a disaster-struck community (?)
- 214: "evaluating quantitatively recovery plans" -> "quantitatively evaluating recovery plans"
- 242: "quick interventions like financial support or food for reconstruction" -> "quick interventions such as financial support for reconstruction or food"
- 275: "Fig." -> Figure
- 346: "while" -> during

**[Authors reply]** We did a language check, including all the issues you highlighted. Regarding specifically your comment:

197: "because socioeconomic recovery is dependent on physical structure" -> is dependent on the recovery of physical structures (?)

The intended meaning here was that physical structures are required for the following socio-economic recovery. We rephrased in such way:

[revised manuscript text omitted]

*5. The current literature does not establish a clear relationship between the response and recovery phases.* Communities show different needs and goals, passing from the short-term response phase to the long-term recovery process. Additionally, from response to recovery, capacities and external contributions may change significantly. Therefore, the transition between the two phases needs to be managed effectively so that stakeholders and decision-makers would not become confused by abrupt changes (Sect. 3.2). Moreover, it could happen for a community to be in the recovery phase from a disaster while in the response phase for another. From a multi-risk perspective environment, the DRM cycle framework has shown limitations in addressing simultaneously response and recovery activities, particularly in multi-risk conditions. In response to these shortcomings, researchers have proposed new frameworks that aim to capture the complexities involved. However, it is worth noting that limited effort has been put to transfer these frameworks into executive guidelines

and plans. (Sect 3.2) to ensure that planning and actions in response and recovery phases should be coordinated to not conflict with one another (Sect. 4.2).

[revised manuscript text omitted]

---

## Author Comment (AC3)

**Review article: current approaches and critical issues in multi-risk recovery planning of urban areas exposed to natural hazards - Soheil Mohammadi et al., 2023**

**Reply to reviewers' comments**

**Reviewer #2 (Robert Sakic Trogrlic)**

Overall comment

My overall comment is that I would like to see the framing of the paper stronger at the very beginning: primarily, why urban areas? In my view, there is a need for a stronger case and I believe this could be easily added. And secondly, the multi-hazard framing needs to be stronger and come much earlier, already in the introduction. Also, how are the findings of your work unique for urban areas? Do they also reflect recovery and reconstruction beyond urban areas?

**[Authors reply]**
Regarding the focus on urban areas, we agree that this aspect was not sufficiently investigated in the introduction. For this reason, we have expanded the problem setting introduction to provide a more comprehensive understanding of the specific features and complexities associated with urban areas in risk management and disaster recovery. Please see the text reported in the answer to **comment C.4.**
.
Regarding the missing multi-risk framing, we provided more context about the multi-risk challenges in disaster recovery and the importance of adopting a multi-risk perspective in this realm, by adding some additional paragraphs in the introduction:
*"When urban systems are affected by complex disaster scenarios, involving potential impact interactions, addressing the recovery process can become more complicated. In that circumstance, the system must be resilient to many types of risks, embracing a multi-risk perspective, which will introduce additional challenges in the decision-making process regarding the recovery (Curt, 2021). Multi-(hazard)-risk, as collectively named by Ward et al. (2022), encompasses all disaster risk assessment and management approaches that consider interactions or interdependencies among different hazards, vulnerabilities, or risks. These approaches are able to better capture complex risk dynamics which are increasingly impacting urban areas worldwide. The interrelationship of multiple hazards and their impacts, as well as the implications of DRM decisions on different economic sectors and regions, and the diverse impact of disaster risk reduction measures on*

*different risks, make recovery in multi-hazard environments challenging (Ward et al., 2022; Hochrainer-Stigler et al., 2023). Among the different types of multi-hazard interaction mechanisms that can lead to a disaster such as compound, triggering, or cascading, (Gill and Malamud, 2014; Marzocchi et al., 2009; Tilloy et al., 2019), the occurrence of consecutive disasters is specifically challenging for the recovery process. Consecutive disasters are two or more disasters that occur in succession and whose direct impacts overlap spatially before the recovery from the prior event is considered complete (de Ruiter et al., 2020). The results of the interaction between the impacts generated by two consecutive hazards depend on the time interval between them, the rate of recovery of the system or the asset, or a combination of them (De Angeli et al., 2022; Marzocchi et al., 2012)."*

Moreover, we supported these concepts by providing a real-world example that highlighted the complex challenges posed by consecutive disasters in recovery:

*"As a real example in the western part of Iran, a devastating earthquake occurred on November 12, 2017, at the Iran-Iraq border, causing the death of at least 630 people (Naserieh et al., 2022). The recovery efforts began by providing temporary shelters to the more than ten thousand affected population (Omarzadeh et al., 2021). Prior to the earthquake, the country was facing a prolonged period of drought, and no one anticipated the possibility of a flood occurring within some months (Yadollahie, 2019). However, from mid-March to April 2019, widespread flash flooding occurred, affecting large areas of Iran, including the regions that were undergoing recovery from the earthquake (Miri et al., 2023). The potential occurrence of a flood was not taken into account, leading to the establishment of temporary shelters alongside canals, which resulted in the flooding of the people residing in those shelters and imposed significant economic impacts on them."*

Regarding the issue of the uniqueness of our findings in relation to urban areas or their broader implications for recovery and reconstruction in general, please refer also to the detailed response provided to **comment C.19**.
In response to comment C.19, we emphasized that while our study primarily focuses on urban areas, the findings and insights that we have obtained can also contribute to a broader understanding of recovery and reconstruction processes in general. The complexities and challenges faced in urban areas, such as diverse stakeholder involvement, interdependencies, and governance issues, are often applicable to recovery and reconstruction efforts in other contexts as well.

More specific comments

**(C.1)** Abstract: In the problem setting, I miss a sentence referring to multi-hazards, before the introduction of the "two-aim"

**[Authors reply]** We agree with this suggestion, and we have included a reference to the "multi-hazard" concept in the introduction in the position you suggested:

*"Post- disasters recovery has been addressed in the literature by different sectoral perspectives and scientific communities. Nevertheless, studies providing holistic approaches to recovery, integrating reconstruction procedures and socio-economic impacts,* **as well as including the additional challenges posed by the effect of complex multiple interacting risks on highly interconnected urban areas,** *are still lacking. Furthermore, recovery has been only marginally explored from a pre-disaster perspective, in terms of planning and actions to increase urban resilience and improve urban systems recoverability"*

**(C.2)** P1 L22 more updated information available?

**[Authors reply]** Thank you for your valuable suggestion. We have replaced the reference with ***Ritchie and Roser (2018)*** to ensure a more updated and accurate citation and provided the appropriate credit.
Additionally, we have modified the term *"more than"* in the sentence to *"almost"* to provide a more accurate representation of the data and analysis presented in the study.

**(C.3)** P1 L27-23 Incomplete sentence

**[Authors reply]**  Thank you for your comment. We have modified the sentence as follows:
*"The origin of the modern resilience theory and its application to natural ecosystems can be traced back to Holling's seminal work in 1973 (Holling, 1973)."*

**(C.4)** General: perhaps use more the term "urban areas" rather than "cities" throughout the paper. What is so specific about urban areas and their hazard and risk scapes that make it particularly important to study? Please elaborate on this in your problem setting.

**[Authors reply]** We have replaced the term *"cities"* with *"urban areas"* in the whole manuscript.
Moreover we elaborated more about the importance and complexity of urban areas for disaster risk management and recovery, including a couple of additional sentences in the introduction. The added paragraph is as follows:

*"Historically, urban areas were often considered safe refuges, shielding inhabitants from the adverse effects of natural hazards. However, a paradigm shift has occurred, acknowledging that cities are now recognized as focal points where disasters and risks converge (Šakić Trogrlić et al., 2018). The rapid expansion of urban areas often necessitates construction in locations that are susceptible to multiple hazards. This is primarily due to*

*limited available land or insufficient time and resources to thoroughly evaluate these areas for their susceptibility to potential interactions between multiple hazards (Jenkins et al., 2023). Furthermore, Urban areas comprise diverse elements, including physical, social, and economic components (Jenkins et al., 2023). These elements collectively contribute to the overall makeup and functioning of urban environments. Moreover, urban areas have a dynamic and unpredictable nature that arises from the interplay between people, activities, institutions, resources, and processes. Enhancing resilience in urban areas can be a complex task, considering the multitude of components, processes, and interactions occurring within and beyond the physical, legal, and virtual boundaries of the urban area (Desouza and Flanery, 2013). An analysis of previous international disaster responses reveals a preference among international humanitarian agencies to provide assistance in rural areas when disasters impact both rural and urban regions (MacRae and Hodgkin, 2016) and when it comes to recovery, urban reconstruction efforts have primarily been undertaken within the scope of national reconstruction programs, receiving limited support from international humanitarian agencies because of the complexities of the recovery process in urban areas (Daly et al., 2017). One significant challenge faced in urban rebuilding endeavors is navigating the intricate web of stakeholders involved in urban environments (Daly et al., 2017). The complexity of urban settings often involves multiple layers of governance, diverse community interests, and various private and public entities, making coordination and decision-making more intricate compared to rural areas."*

**(C.5)** P3 L82-83 New research directions will not "enable" stakeholders to improve decision-making per se and by themselves. Therefore, this is an overstatement, please rephrase.

**[Authors reply]** We have modified the sentence as follows:
*"The ultimate objective is to propose new research directions that can inform stakeholders' decision-making processes and optimize their investments in the pre-disaster phase. This contribution aims to enhance the recoverability of urban areas."*

**(C.6)** P3 L86 Perhaps this should be a separate heading and section focusing on the details of the study approach. The Introduction is currently way too long and at times difficult to follow.

**[Authors reply]** Thank you for this valuable suggestion. We created a separate section (Sect. 2, Methodology) where we moved the text originally reported in L.86-131. This change has allowed us to provide more visibility to the methodology on one side, but also to lighten the introduction. This last aspect has allowed us to strengthen the multi-risk aspects of the introduction by adding some dedicated paragraphs *(see*

**also reply to comment C.7**). Moreover, as suggested in comment **C.10**, we also included in the new section 'Methodology' a figure (Fig.1), where we graphically depicted the relationship between the identified Issues and the guiding Research Questions. The figure is attached to this document as an annex.

**(C.7)** General: in the Introduction, I miss the definition of multi-hazards and multi-risks and why are these crucial to be studied in the context of urban recovery and urban resilience. Perhaps also include a real-world example of that shows the completely of MH recovery? This is currently way too late in the paper.

**[Authors reply]** We provided more context about the multi-risk challenges in disaster recovery and the importance of adopting a multi-risk perspective in this realm, by adding some additional paragraphs in the introduction:

*"When urban systems are affected by complex disaster scenarios, involving potential impact interactions, addressing the recovery process can become more complicated. In that circumstance, the system must be resilient to many types of risks, embracing a multi-risk perspective, which will introduce additional challenges in the decision-making process regarding the recovery (Curt, 2021). Multi-(hazard)-risk, as collectively named by Ward et al. (2022), encompasses all disaster risk assessment and management approaches that consider interactions or interdependencies among different hazards, vulnerabilities, or risks. These approaches are able to better capture complex risk dynamics which are increasingly impacting urban areas worldwide. The interrelationship of multiple hazards and their impacts, as well as the implications of DRM decisions on different economic sectors and regions, and the diverse impact of disaster risk reduction measures on different risks, make recovery in multi-hazard environments challenging (Ward et al., 2022; Hochrainer-Stigler et al., 2023). Among the different types of multi-hazard interaction mechanisms that can lead to a disaster such as compound, triggering, or cascading, (Gill and Malamud, 2014; Marzocchi et al., 2009; Tilloy et al., 2019), the occurrence of consecutive disasters is specifically challenging for the recovery process. Consecutive disasters are two or more disasters that occur in succession and whose direct impacts overlap spatially before the recovery from the prior event is considered complete (de Ruiter et al., 2020). The results of the interaction between the impacts generated by two consecutive hazards depend on the time interval between them, the rate of recovery of the system or the asset, or a combination of them (De Angeli et al., 2022; Marzocchi et al., 2012)."*

Moreover, we supported these concepts by providing a real-world example that highlighted the complex challenges posed by consecutive disasters in recovery:

*"As a real example in the western part of Iran, a devastating earthquake occurred on November 12, 2017, at the Iran-Iraq border, causing the death of at least 630 people (Naserieh et al., 2022). The recovery efforts began by providing temporary shelters to the more than ten thousand affected population (Omarzadeh et al., 2021). Prior to the earthquake, the country was facing a prolonged period of drought, and no one anticipated*

*the possibility of a flood occurring within some months (Yadollahie, 2019). However, from mid-March to April 2019, widespread flash flooding occurred, affecting large areas of Iran, including the regions that were undergoing recovery from the earthquake (Miri et al., 2023). The potential occurrence of a flood was not taken into account, leading to the establishment of temporary shelters alongside canals, which resulted in the flooding of the people residing in those shelters and imposed significant economic impacts on them."*

The multi-risk concept has been also strengthened in other sessions on the manuscript, by adding some dedicated paragraphs inside the discussion of Issues 1, 2, and 3, discussing more in detail:
- the difficulty of keeping DRM phases separate when dealing with multi-risk and emerging DRM models alternative to the traditional cyclic approach (Sect. 2.2)
- the exploitation of the build-back-better concept from a multi-risk perspective (Sect. 2.3)
- how multi-hazard conditions can exacerbate the communities' dilemma in balancing the need to enhance resilience against future disasters and the desire to quickly rebuild their houses and livelihoods (Sect. 3.1)
- multi-risk multi-sectoral challenges related to interdependencies among urban system components (Sect. 4)

As a consequence of this spread of multi-risk related criticalities along the manuscript, we also reorganized more effectively the section originally devoted to multi-risk (Sect. 3.2). More specifically we focused it more on the recovery requirements seen from a multi-risk perspective and we consequently renamed it as "Multi-hazard risk recovery requirements". Furthermore, we moved some of the paragraphs into the Introduction, and added some paragraphs covering the following aspects:

- multi-risk prioritization in recovery planning
- asynergies of DRR measures implemented during reconstruction and recovery
- impact of multi-risk scenarios on the allocation of resources for recovery activities

**(C.8)** General: The DRM cycle is highly criticised by some researchers as the phases are not easily distinguishable and they overlap in practice. Would be useful if you would make sure this literature is included and reflected on.

**[Authors reply]** Thank you for your comment. We have carefully reviewed the initial paragraph of section 2.2 and made the necessary modifications. The revised paragraph now includes sentences addressing the literature on DRM criticism.

The updated paragraph reads as follows:

*"The DRM cycle is widely recognized in the global DRM community as a framework for managing various types of disasters, both natural and anthropogenic (Coetzee and Van, 2012). It is characterized by separate and sequential phases with varying durations and actions. While phases' number and their naming vary in the literature, the following main ones (before, during, and after the event) can be identified: preparedness and mitigation, response, and recovery. However, the current DRM cycle falls short in effectively addressing the complexities of multi-hazard risk scenarios. The DRM cycle, characterized by separate and sequential phases, fails to adequately capture the dynamics and interaction of these multiple hazards particularly those involving both sudden-onset (e.g., earthquakes, flash floods) and slow-onset hazards (e.g., pandemics, droughts, conflicts) (Terzi et al., 2022). Consequently, numerous authors have proposed alternative frameworks for DRM, challenging the current circular representation (Bosher et al., 2021; Staupe-Delgado, 2019; Terzi et al., 2022). Despite the aforementioned considerations, the conventional DRM cycle continues to prevail as the predominant discourse within the realms of decision makers, practitioners, and the academic community focused on disaster risk and it is used as a reference for the current research."*

**(C.9)** P4 L113 Was there a combination of keywords used and if yes, what combinations?

**[Authors reply]** We did not use keywords in a combined way since some of them were already composed of more than a single word.

**(C.10)** General: I strongly advise creating a figure that will represent your literature searcher and selections and a series of research issues mapped against your objectives. It will aid the clear overview of the paper and make it easier for your reader.

**[Authors reply]** We added a figure (Fig.1 in Sect.2, Methodology), where we graphically depicted the relationship between the identified Issues and the guiding Research Questions. The figure is attached to this document as an annex.

**(C.11)** P5 L151: Nice distinction, however, a dated reference. I am wondering if there is anything more up to date?

**[Authors reply]** We appreciate your suggestion to include an additional, up-to-date reference in the manuscript. We have now included the reference to ***Ryan et al. (2016)*** to provide a more comprehensive and current perspective on the topic.

**(C.12)** P7 L194: Perhaps add "recovery" before "socio-economic activities"

**[Authors reply]**  We have added the term according to your suggestion.

**(C.13)** Figure 1: Great figure and very clear! Was this based on some inputs from a literature as there are not references in the figure? Fine if not, but wondering about the description of the figure in the text: I would expect some references as it is a review paper?

**[Authors reply]** Thank you for your comment. We apologize for any confusion regarding the origin of the figure. We would like to clarify that the figure was developed by the authors based on the existing literature in the field. To ensure proper attribution, we have added references to the description of the figure in the text. This allows readers to understand the sources that informed our conceptualization and representation. The updated description paragraphs reads as follows:

*"In an ideal recovery process, the intensity of physical reconstruction and socio-economic recovery activity remains at their maximum level during the recovery time to maximise the functional recovery zone. However, this would be very challenging. As will be discussed in Sect. 2.2, resource allocation and concentration on the disaster area will not remain constant during the whole recovery period. External support would decline, and the intensity of reconstruction activities could decrease (Choi et al., 2019). Nevertheless, because socioeconomic recovery is dependent on physical structure (Barakat and Zyck, 2011; Mitsova et al., 2019) , the intensity of related recovery activities may be raised with the repair of some of the damaged facilities, while at the initial stage of the recovery, it might not be so high due to the damaged structures and infrastructures.*
*As shown in panel (b) of Fig. 1, by increasing the intensity of socio-economic recovery activities, the common zone (functional recovery) could be preserved or even increased during the recovery period. Furthermore, the intensity of socio-economic recovery activities could be increased without external support in a disaster-struck community (Alifa and Nugroho, 2019). These cooperatively evolving activities enable people to take part in the restoration of their communities independently (Nigg, 1995; Talbot et al., 2020; Perce, 2007). For instance, as more enterprises of all sizes become involved in the economy, people will be more capable of actively participating in the economic recovery of their community (Freeman, 2004). Setting this balance between physical reconstruction and socioeconomic recovery would be possible if the disaster area needs assessment (see Sect. 4.13) is considered."*

**(C.14)** P9 L229: The fact that the phases should be separate. This could be problematic – please see my comment above on the criticism on the DRM cycle.

**[Authors reply]** We agree that phases cannot always be easily separated and that in some cases it can be necessary to adopt a more fuzzy approach compared to the traditional DRM cycle with consecutive, well-defined, and separate phases. Nevertheless, understanding the relationship between them still remains crucial, specifically considering the relationship between short-term response actions and long-term recovery efforts (see Sect. 2.2, L. 230-233: *"However, it should be noted that experience indicates that addressing the short-term requirements of affected populations during the response phase has an influence on meeting the needs of the population during long-term recovery and addressing these two types of needs should be done in an integrated way (Garnett and Moore, 2010)"* )

Nevertheless, we agree that the sentence reported at L229 was misleading and we eliminated it from the manuscript.

Moreover, in line with this perspective, we also changed the title of one of the ten final key challenges (Challenge number 5) discussed in Sect. 5., replacing the word "distinction" with "relationship":

*5. The current literature does not establish a clear relationship between the response and recovery phases.*

**(C.15)** General: The paper has a lot of very long paragraphs that sometimes make the reading a bit challenging. I suggest editing this throughout the paper, shortening and separating thoughts per paragraphs where possible.

**[Authors reply]** To improve the flow of the text, we checked the overall manuscript and reorganized the sentences into coherent paragraphs, avoiding having too small or too long paragraphs.

**(C.16)** P13 L350-355 Really interesting! Any further explanation on why? Perhaps do reflect on the literature on the role of politics and electoral cycles- even with a reference or two.

**[Authors reply]** Thank you for your suggestion. We have carefully considered your comment and have added the following paragraph to address the issue of the electoral cycle in decision-making regarding the recovery process:

*"During an electoral cycle, politicians often display a tendency to prioritize physical reconstruction efforts over socio-economic recovery. This preference entails a stronger focus on rebuilding infrastructure and physical structures, driven by political motivations*

*to demonstrate tangible progress and garner public support (Masiero and Santarossa, 2021)."*

We believe that this addition enhances the discussion by addressing the role of the electoral cycle in decision-making during the recovery process. Thank you for bringing this important aspect to our attention.

**(C.17)** P13 L366 on community involvement – I truly enjoyed reading this part of the paper. However, what I miss is mentioning the need for community involvement in pre-disaster planning, also getting input and community priorities into the recovery and reconstruction plans.

**[Authors reply]** Thank you for your positive feedback. We appreciate your recognition of the importance of community involvement in recovery planning during the pre-disaster period. We agree that this is a crucial aspect that deserves attention. We have made the following additions to highlight its significance, and to exemplify the significance of community involvement in the pre-disaster we present a real-life case study of the aftermath of Hurricane María in Puerto Rico.

*"Pre-disaster community involvement in recovery planning and social organization plays a pivotal role in enhancing the community's capability to leverage its capacities, particularly its social capacity, to effectively address challenges in the aftermath of a disaster and actively engage in the recovery process. Social organization through the involvement of different stakeholders in pre-disaster planning significantly contributes to the successful implementation of recovery plans and facilitates the realization of post-disaster recovery efforts (Delilah Roque et al., 2020).*

*An illustrative example of the significance of social organizations can be observed in the aftermath of Hurricane María in Puerto Rico. During this time, social organizations played a crucial role in coordinating and addressing the comprehensive needs of the community. Despite the island-wide power outage, these organizations became vital hubs where community members could come together to organize, support one another, and address the challenges faced. Their pre-disaster planning efforts and established networks allowed for effective coping mechanisms and resource allocation, even in the absence of immediate access to personal finances held in banks (Delilah Roque et al., 2020)."*

**(C.18)** Table 3: it mentions seven publications in the table title, while four are listed.

**[Authors reply]** Thank you for pointing out the mistake in reporting the number of publications in the manuscript text. We apologize for the error and have made the necessary correction.

**(C.19)** Section 5: Which of these are specific to urban areas and which in general to recovery and reconstruction?

**[Authors reply]** The present research addresses the recovery issues associated with multi-hazard events in urban environments, taking into account the complexity of urban systems, their diverse goals and perspectives, and the heterogeneous stakeholder involvement. The final section of the manuscript integrates and summarizes the challenges discussed throughout the paper. This serves as a background for identifying future research needs in the field, providing the key findings and take-home messages of the current study. As a result, it can be claimed that the urban system could be affected by all of the challenges outlined, and all of the challenges are associated with urban areas. Nevertheless, it is important to note that while all the challenges outlined primarily pertain to urban areas, some of them could also be applicable to the recovery processes in other contexts, such as rural settlements. Specifically, certain challenges directly highlight urban aspects, such as challenge number 7, which emphasizes the need to view disasters as opportunities to enhance the resilience of *"urban systems"*. Additionally, challenge number 10 underscores the significance of understanding the socioeconomics and interdependencies among various "*urban assets"* or sectors within the "*urban system"*. In challenge number 2, the term *"urban area"* is added to emphasize the heterogeneity of the stakeholder groups involved in "*urban systems"*.

We highlighted this concept in the conclusions adding the following sentence:

*"The outcomes of this critical literature can set the basis to outline the key research directions in the field of multi-risk disaster recovery and urban resilience. Nevertheless, it is important to note that while all the challenges outlined primarily pertain to urban areas, some of them could also be applicable to the recovery processes in other contexts, such as rural settlements."*

**Annex 1**

[Figure]

**Figure 1. Graphical representation of the methodological steps implemented to perform the critical literature review on multi-risk recovery planning, highlighting the relationship between the guiding Research Questions and the multi-risk recovery planning Issues**

---

## Referee Report (RR1)

egusphere-2023-504    Submitted on 20 Mar 2023
**Review article: Current approaches and critical issues in multi-risk recovery planning of urban areas exposed to natural hazards**
Soheil Mohammadi, Silvia De Angeli, Giorgio Boni, Francesca Pirlone, and Serena Cattari

Dear authors,

Thank you very much for your efforts and congratulations for tackling the comments in such an extensive and detailed way. In my view, this has now greatly improved the manuscript. As mentioned previously, I believe this paper is a great addition to the growing field of understanding multi-hazard risks and associated management practices.

Please see some minor comments below:

- P1 L8-10 Suggestion to separate this into two sentences
- P1 L11 ''Improve urban system recoverability'' – In my view, this is redundant, please reconsider.
- In the Abstract, and later in the paper, you mix multi-risk and multi-hazard risk. Please ensure the consistency, especially as later on you refer to Ward et al 2022 paper.
- P1 L28 ''hazard mitigation'' it is not clear what this refers to and entails
- P2 L22 ''safe refuges'' to be changes to ''safe refuge'' or ''safe heaven.''
- P2 L60 change ''composed by'' to ''composed of''
- P2 L64 Not just rapid expansion, but also unplanned and not risk informed. Please see Cremen et al. (2023) A state-of-the-art decision-support environment for risk-sensitive and pro-poor urban planning and design in Tomorrow's cities
- P2 L70 Meerow et al (2016), please add page number as this is a direct quote
- P3 L124-126 A lot of research is quantifying the benefit rations of investing in preparedness. Perhaps worthwhile reflecting on this research.
- P4 L218 Please rephrase ''provide some hints'' when referring to the methodology section
- Figure 1: Many thanks for making it. I suggest adding some keywords below ''Keywords + databases'' and an indication of a number of papers.
- Table 1: If you see as important, perhaps add a column that will add ''recovery as a desired outcome'' and ''recovery as a process'', to align text and definitions
- P11 L470 Reflect on the overlap of DRM phases (see Twigg, J., 2015. Disaster Risk Reduction- Good Practice Review 9. 2nd ed. London, UK: Overseas Development Institute.)
- P12 L508-512 Your paragraphs are still very long which makes it at times challenging for the reader. Perhaps in places like this you could simplify through bullet pointing to increase the readability.
- P12 L512 Naturally, here is a new paragraph?
- P12 L517 Same as the previous comment.
- P13 L557 Please define functionality.
- P15 L643 ''Emergency phase'' or ''Response phase''? Response has been used previously. Please ensure consistency of terms throughout the paper.
- Table 2: You mention 8 publications but have only 2 in the table?
- P22 L900-902 Please rephrase the sentence.
- P23 Too long – an example where the text could be cut significantly. Perhaps some of the critique can be integrated in Table 3 and text shortened.
- Figure 5: instead of ''time frame'' please add ''time frames''

---

## Author Response (AR2)

**Review article: current approaches and critical issues in multi-risk recovery planning of urban areas exposed to natural hazards - Soheil Mohammadi et al., 2023**

**Reply to reviewers' comments**

Dear Editor and Reviewers

Thank you for your valuable feedback. We appreciate the constructive comments and guidance provided.

We have addressed the minor revisions suggested and provided a detailed reply to the reviewers' comments. Moreover, we incorporated language-related improvements in the revised version of the manuscript. We have included a track changes document that highlights the modifications made between the old and new versions.

Moreover, in consideration of the modifications in the manuscript based on the reviewers' suggestions, we have also provided a revised version of the supplementary material. Some minor changes have been made in the section names to align with the modifications in the manuscript.

We appreciate your time and consideration of our work and look forward to any further guidance you may provide.

Soheil Mohammadi and Co-authors

**Reviewer #1 (Marleen de Ruiter)**

I would like to thank the authors for their very elaborate response. They have done an excellent job and the manuscript has improved greatly. The manuscript now flows very logically and its aim is very clear. Moreover, the supplementary material is very clearly formatted.

Below I have some very minor and some language-related suggestions:

- L. 54: elaboration of

- L. 195: "...challenge recovery in multi-hazard environments."

- L. 202: "As a real example in the western part of Iran, a devastating earthquake occurred on November 12, 2017, at the Iran-Iraq border, causing the death of at least 630 people (Naserieh et al., 2022)."

o It is very nice that the authors included this example

o Please adjust: "real world" instead of real.

o Also, it may be interesting for the reader to know the magnitude of the earthquake; this can maybe be added to the sentence? ("…a devastating earthquake of magnitude XX…")

- L. 204: "…the more than ten thousand affected population…" -> "…the more than ten thousand affected people…"

- Fig 1 is an excellent addition to the paper (as suggested by review 2).

- L. 376: "The fundamental distinction between the two definitions of 'recovery' and 'reconstruction' provided by UNDRR"

o This source requires a year of publication (I assume 2020)

- L. 475: "…do not adequately…" ->The cycle (…) "does not"

- L. 603: "In the field of economic studies…" -> "In the field of economics…"

- L. 1006: "…divided into three time frame.." -> "…divided into three time frames.."

- L. 1159: "…In this final session.." -> section

**[Authors reply]:**

Dear Dr. De Ruiter,
Thank you for your thoughtful feedback and acknowledgment of our efforts to enhance the manuscript. We appreciate your positive comments on the improved logical flow and clarity of purpose. We also take note of your minor language-related suggestions and have carefully addressed and incorporated them to further enhance the quality of our work.

**Reviewer #2 (Robert Sakic Trogrlic)**

Overall comment

Dear authors,

Thank you very much for your efforts and congratulations for tackling the comments in such an extensive and detailed way. In my view, this has now greatly improved the manuscript. As mentioned previously, I believe this paper is a great

addition to the growing field of understanding multi-hazard risks and associated management practices.

**[Authors reply]**

Dear Dr. Trogrlic

Thank you sincerely for your kind words and appreciation. We are delighted to hear that our efforts in addressing the comments have significantly improved the manuscript. Your positive assessment and recognition of the paper's contribution to the field of multi-hazard risk understanding and management practices are truly motivating. We greatly appreciate your valuable feedback.

More specific comments

**(C.1)** P1 L8-10 Suggestion to separate this into two sentences

**[Authors reply]** We agree with this suggestion, and we have separated the sentence into two sentence to improve the readability:

*"Nevertheless, studies providing holistic approaches to recovery, integrating reconstruction procedures and socio-economic impacts, are still lacking. Additionally, there is a gap in disaster recovery research addressing the additional challenges posed by the effect of complex, multiple, and interacting risks on highly interconnected urban areas."*

**(C.2)** P1 L11 "Improve urban system recoverability" – In my view, this is redundant, please reconsider.

**[Authors reply]** Thank you for your suggestion. We have changed the sentence and remove the redundant part.

**(C.3)** In the Abstract, and later in the paper, you mix multi-risk and multi-hazard risk. Please ensure the consistency, especially as later on you refer to Ward et al 2022 paper.

**[Authors reply]** Thank you for your comment. We have revised the terminology in the Abstract. Furthermore, we have thoroughly reviewed the entire paper to ensure consistency and made corresponding modifications where needed.

**(C.4-5)** P2 L22 "safe refuges" to be changes to "safe refuge" or "safe heaven." P2 L60 change "composed by" to "composed of"

**[Authors reply]** We agree with your comment. We have modified the terms.

**(C.6)** P2 L64 Not just rapid expansion, but also unplanned and not risk informed. Please see Cremen et al. (2023) A state-of-the-art decision-support environment for risk-sensitive and pro-poor urban planning and design in Tomorrow's cities.

**[Authors reply]** Thank you very much for your invaluable comment. We have added the *"unplanned and not risk informed"* to the sentence:

*" In fact, the rapid, unplanned, and not risk-informed expansion of urban areas often necessitates construction in locations that are susceptible to multiple hazards (Cremen et al., 2023)"*

**(C.7)** P2 L70 Meerow et al (2016), please add page number as this is a direct quote

**[Authors reply]** Thank you for noticing that. We carefully checked it and we found that it is not required according to the "English guidelines and house standards" section in the NHESS Submission instructions.

**(C.8)** P3 L124-126 A lot of research is quantifying the benefit rations of investing in preparedness. Perhaps worthwhile reflecting on this research.

**[Authors reply]** Thank you for your comment. We have considered literature that quantifies the benefit ratios of investing in preparedness, and we have modified the sentence as follows:

*"The resilience concept can be reflected in preparedness, response, recovery, and adaptation actions, depending on the temporal domain. Even though numerous studies have quantified the benefits of investing in preparedness by comparing potential damage and preparedness costs (Goldschmidt and Kumar, 2019; David R. et al., 2009; Kousky et al., 2019; Heo and Heo, 2019), there is still a notable absence of research that systematically evaluates the relationships and implications of different actions on one another in resilience-based urban planning are still absent (Rus et al., 2018)."*

**(C.9)** Figure 1: Many thanks for making it. I suggest adding some keywords below "Keywords + databases" and an indication of a number of papers.

**[Authors reply]** Thank you for your previous constructive comment. The suggestion to include the Figure was also appreciated by the other reviewer, and it has enhanced the manuscript. We have made modifications to the figure, and it now appears as follows:

[Figure]

**(C.10)** • Table 1: If you see as important, perhaps add a column that will add "recovery as a desired outcome" and "recovery as a process", to align text and definitions.

**[Authors reply]** Thank you for your comment. It is worth noting. However, since only one of the definitions in the table considers recovery as a desired outcome, we have opted to address this by adding a sentence to the text, rather than incorporating an additional column into the table.
*"In Table 1, all definitions, except the one provided by Quarantelli (1989), conceptualize recovery as a process rather than a desired outcome. "*

**(C.11)** P11 L470 Reflect on the overlap of DRM phases (see Twigg, J., 2015. Disaster Risk Reduction- Good Practice Review 9. 2nd ed. London, UK: Overseas Development Institute.)

**[Authors reply]** We appreciate your suggestion to reflect on the mentioned research. We have now included the reference to **Twigg, J. (2015)** and added a sentence as follow.
*"However, the current understanding of recovery recognizes it as an ongoing, long-term process that can start simultaneously with the response phase, and the developmental recovery activities are extended alongside the mitigation phase, leading to the overlap of different phases in practice (Twigg, 2015). "*

**(C.12)** P12 L508-512 Your paragraphs are still very long which makes it at times challenging for the reader. Perhaps in places like this you could simplify through bullet pointing to increase the readability.

**[Authors reply]** We have added bullet point according to your suggestion.

**(C.13-14)** P12 L512 Naturally, here is a new paragraph? P12 L517 Same as the previous comment.

**[Authors reply]** Thank you for your comment. We have divided the text into two paragraphs

**(C.15)** P13 L557 Please define functionality.

**[Authors reply]** Thank you for your comment. We have defined the functionality as follow:

*"In this study, the urban system functionality denotes the effective and interdependent operation of infrastructure, services, and socio-economic activities within a city to meet the needs of its population and safeguard them against potential hazards."*

**(C.16)** · P15 L643 "Emergency phase" or "Response phase"? Response has been used previously. Please ensure consistency of terms throughout the paper.

**[Authors reply]** Thank you for noticing that. We have gone through the manuscript and tried to be consistent concerning the "Response phase".

**(C.17)** Table 2: You mention 8 publications but have only 2 in the table?

**[Authors reply]** Thank you for noticing that. We have included 6 publication and modified the table caption accordingly.

**(C.18)** P22 L900-902 Please rephrase the sentence.

**[Authors reply]** Thank you for your suggestion. We rephrase the sentence as follow:

*"However, there is often not enough focus on land-use planning for flood mitigation and evacuation modelling or in general flood disaster risk management"*

**(C.19)** P23 Too long – an example where the text could be cut significantly. Perhaps some of the critique can be integrated in Table 3 and text shortened.

**[Authors reply]** Thank you for your comment. While we appreciate the suggestion to integrate some of the critique into Table 3 and shorten the text on page 23, we believe that It's important to note that the extended content in this section is dedicated to thoroughly discussing current approaches in multi-risk recovery, including their respective advantages and disadvantages.

Moreover, please consider that the table in the text represents a condensed version of the complete table found in the supplementary material. Each of the 4 mentioned publication in the table represent a specific approach, and the detailed nuances of these approaches might not be possible if integrate the content into the table.

**(C.20)** Figure 5: instead of "time frame" please add "time frames"

**[Authors reply]** Thank you for your comment. We modified the Figure 5.